# CAdam: Confidence-Based Optimization for Online Learning

## Abstract

Modern recommendation systems frequently employ online learning to dynamically update their models with freshly collected data. The most commonly used optimizer for updating neural networks in these contexts is the Adam optimizer, which integrates momentum ($m_t$) and adaptive learning rate ($v_t$). However, the volatile nature of online learning data, characterized by its frequent distribution shifts and presence of noise, poses significant challenges to Adam's standard optimization process: (1) Adam may use outdated momentum and the average of squared gradients, resulting in slower adaptation to distribution changes, and (2) Adam's performance is adversely affected by data noise. To mitigate these issues, we introduce CAdam, a confidence-based optimization strategy that assesses the consistency between the momentum and the gradient for each parameter dimension before deciding on updates. If momentum and gradient are in sync, CAdam proceeds with parameter updates according to Adam's original formulation; if not, it temporarily withholds updates and monitors potential shifts in data distribution in subsequent iterations. This method allows CAdam to distinguish between the true distributional shifts and mere noise, and to adapt more quickly to new data distributions. In various settings with distribution shift or noise, our experiments demonstrate that CAdam surpasses other well-known optimizers, including the original Adam. Furthermore, in large-scale A/B testing within a live recommendation system, CAdam significantly enhances model performance compared to Adam, leading to substantial increases in the system's gross merchandise volume (GMV).

## 1 Introduction

Online learning has become a key paradigm in modern machine learning systems, especially in real-time applications such as online advertising and recommendation platforms (Ko et al., 2022). In these settings, models are updated continuously using newly arrived data, allowing them to quickly adapt to shifting user behavior. Unlike traditional offline training, which operates on fixed datasets, online learning must handle streaming inputs that are often non-stationary and noisy. A typical online learning system operates sequentially: at each time step, it observes an input $x$—comprising user and item features such as age, gender, and behavioral history—and predicts an outcome $f_w(x)$ parameterized by $w$, which is trained to match the observed label $y$, such as a click or purchase.

Among optimizers, Adam (Kingma and Ba, 2015) has become a default choice across a wide range of machine learning tasks due to its fast convergence and robustness to gradient scaling. By combining momentum with coordinate-wise adaptive learning rates, Adam has demonstrated strong empirical performance in computer vision (Alexey, 2020), natural language processing (Vaswani, 2017), and reinforcement learning (Schulman et al., 2017). Its plug-and-play nature and relatively low tuning cost also make it attractive for production.

However, in online learning environments, Adam faces two fundamental challenges that limit its effectiveness. First, the data distribution $P_t(x)$ can shift rapidly due to changing user cohorts and behavioral patterns. For example, elderly users may dominate the platform in the morning, while working professionals are more active in the evening. Although the underlying user preference function $\hat{f}(x)$ may remain relatively stable, the input $x$ varies significantly over time. Adam accumulates momentum and second-moment estimates based on historical inputs $x_{\text{old}}$, which may no longer reflect the current distribution. This misalignment causes updates to follow outdated directions, as illustrated

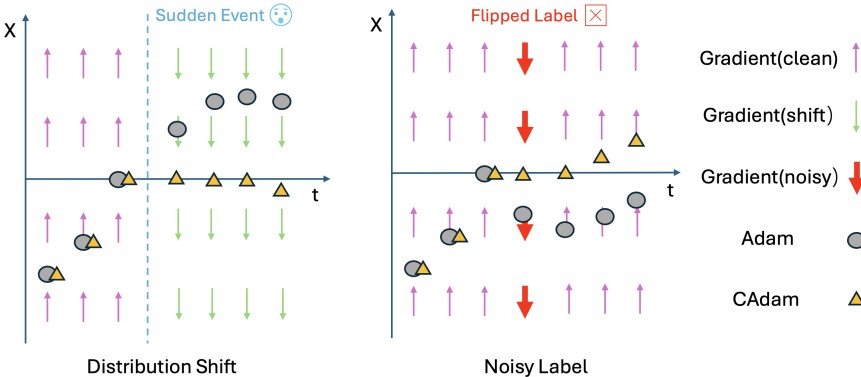

Figure 1: Illustration of the motivation behind CAdam. **Left:** Under a distribution shift, Adam continues updating in outdated directions due to stale momentum, while CAdam adapts quickly by pausing misaligned updates. **Right:** When encountering a noisy label, Adam blindly follows the misleading gradient, but CAdam stops updating by checking gradient-momentum alignment.

in Figure 1 (Left), where the optimizer continues to move toward stale optima even after the true target has shifted. Such behavior becomes increasingly problematic under large distribution shifts—a phenomenon further reflected in various evaluation settings (e.g., Figure 2, 4, 5).

Second, online feedback is often noisy. Users may click on irrelevant ads by mistake (false positives) or overlook relevant ones due to distraction or fatigue (false negatives), introducing randomness into the supervision signals (Wang et al., 2021). Since Adam treats all samples equally, it cannot distinguish between clean and noisy updates—causing noisy examples to influence the optimization trajectory as visualized in Figure 1 (Right). In high-variance environments (Yang et al., 2023), this sensitivity to noise can lead to degraded stability and convergence behavior, as reflected in later results (e.g., Figure 3, 5).

To address these issues, we propose **Confidence Adaptive Moment Estimation (CAdam)**, a simple yet effective variant of Adam designed for online learning. The core idea is to introduce a confidence-aware mechanism that assesses the reliability of each gradient update by comparing the current gradient $g_t$ and the historical momentum $m_t$. This selective update strategy allows CAdam to avoid blindly following outdated momentum and to prevent noisy gradients from disrupting the learning process. Importantly, CAdam requires no additional hyperparameters and can be used as a drop-in replacement for Adam in existing systems. We further provide a theoretical analysis showing that CAdam retains the same convergence rate as Adam under mild assumptions, making it a reliable alternative for real-world online learning applications.

Our contributions can be summarized as follows:

1. We propose CAdam, a confidence-aware optimizer tailored for online learning scenarios. It enhances Adam by dynamically adjusting its updates to account for both label noise and distribution shifts.

2. Through comprehensive experiments spanning numerical optimization, image classification with distribution shift or noise, and recommendation benchmarks, we demonstrate that CAdam consistently outperforms other optimizers, including Adam, showing stronger robustness to noise and faster adaptation to distributional changes.

3. We validate CAdam in a production-scale recommendation system via large-scale online A/B testing. Notably, CAdam has been running in production across 16 online scenarios, serving millions of users continuously for over nine months.

## 2 DETAILS OF CADAM OPTIMIZER

**Notations** We use the following notations for the CAdam optimizer:

- $f(\theta) \in \mathbb{R}, \theta \in \mathbb{R}^d$: Stochastic objective function to minimize, where $\theta$ is the parameter in $\mathbb{R}^d$.

- $g_t$: Gradient at step $t$, defined as $g_t = \nabla_\theta f_t(\theta_{t-1})$.

- $m_t$: Exponential moving average (EMA) of $g_t$, computed as $m_t = \beta_1 \cdot m_{t-1} + (1 - \beta_1) \cdot g_t$.

- $v_t$: EMA of the squared gradients, given by $v_t = \beta_2 \cdot v_{t-1} + (1 - \beta_2) \cdot g_t^2$.

- $\hat{m}_t, \hat{v}_t$: Bias-corrected estimates of $m_t$ and $v_t$, where $\hat{m}_t = \frac{m_t}{1-\beta_1^t}$ and $\hat{v}_t = \frac{v_t}{1-\beta_2^t}$.

- $\alpha, \epsilon$: Learning rate $\alpha$, typically set to $10^{-3}$, and $\epsilon$, a small constant to prevent division by zero.

- $\beta_1, \beta_2$: Smoothing parameters, typically $\beta_1 = 0.9$, $\beta_2 = 0.999$.

- $\theta_t$: Parameter vector at step $t$.

- $\theta_0$: Initial parameter vector.

---

**Algorithm 1** Confidence Adaptive Moment Estimation (CAdam)

---

1:  $m_0 \leftarrow 0$, $v_0 \leftarrow 0$, $\hat{v}_{\max,0} \leftarrow 0$, $t \leftarrow 0$, $\theta_t = \theta_0$
2:  **while** $\theta_t$ not converged **do**
3:    $t \leftarrow t + 1$
4:    $g_t \leftarrow \nabla_\theta f_t(\theta_{t-1})$
5:    $m_t \leftarrow \beta_1 \cdot m_{t-1} + (1 - \beta_1) \cdot g_t$
6:    $v_t \leftarrow \beta_2 \cdot v_{t-1} + (1 - \beta_2) \cdot g_t^2$
7:    $\hat{m}_t \leftarrow m_t/(1 - \beta_1^t)$
8:    $\hat{v}_t \leftarrow v_t/(1 - \beta_2^t)$
9:    **if** AMSGrad **then**
10:       $\hat{v}_{\max,t} \leftarrow \max(\hat{v}_{\max,t-1}, \hat{v}_t)$
11:    **else**
12:       $\hat{v}_{\max,t} \leftarrow \hat{v}_t$
13:    **end if**
14:    $\hat{m}_t \leftarrow \hat{m}_t \odot \mathbb{I}(m_t \odot g_t > 0)$ // *Element-wise mask out elements where $m_t^i \cdot g_t^i \leq 0$*
15:    $\theta_t \leftarrow \theta_{t-1} - \alpha \cdot \hat{m}_t/(\sqrt{\hat{v}_{\max,t}} + \epsilon)$
16: **end while**
17: **return** $\theta_t$

---

**Comparison with Adam** CAdam (Algorithm 1) and Adam both rely on the first and second moments of gradients to adapt learning rates. The key difference is that CAdam introduces the alignment between the momentum and the current gradient as a confidence metric to address two major challenges in online learning: distribution shifts and label noise, as shown in Figure 1.

In Adam, the update direction is determined by the exponential moving average of gradients $m_t$ and squared gradients $v_t$. This works well under stable data distributions, where $m_t$ tracks the general optimization direction. However, when the distribution changes, $m_t$ may lag behind and point in outdated directions. Adam will continue to update parameters using this stale momentum until $m_t$ gradually realigns with the new gradient direction, which can degrade performance during transition periods. Moreover, when facing noisy samples, Adam treats them the same as clean data—blindly applying updates—potentially amplifying the noise due to increased gradient variance.

CAdam addresses these issues by checking whether the current gradient $g_t$ and momentum $m_t$ are aligned before updating. If they point in the same direction, CAdam proceeds with the standard update using $m_t/\sqrt{v_t}$. If they point in opposite directions, CAdam temporarily pauses the update for that parameter. This mechanism enables CAdam to distinguish between two cases: (1) If the direction of $g_t$ remains consistent in subsequent steps, it likely reflects a real distribution shift. By not updating prematurely, CAdam avoids reinforcing outdated directions and allows the momentum to decay naturally, eventually following the new gradient trend. (2) If the disagreement disappears quickly—because the gradient direction reverts or fluctuates randomly—it is likely due to noise. In this case, CAdam resumes normal updates in the next steps, effectively filtering out one-off noisy signals without affecting future learning.

Additionally, CAdam supports an AMSGrad-style update (Reddi et al., 2018) as described in Algorithm 1.

# 3 EXPERIMENT

## 3.1 NUMERICAL EXPERIMENT

To intuitively demonstrate the advantages of CAdam in dynamic and noisy environments, we present two numerical experiments. In the first setting (Figure 2), we simulate distribution shifts by continuously changing the optimal solution over time. The figure shows that CAdam tracks the moving minima more accurately than Adam by avoiding misleading updates caused by outdated momentum.

In the second setting (Figure 3), we optimize over a fixed two-dimensional landscape but introduce noise by randomly modifying the function value: with 0.5 probability, $f(x, y)$ is scaled by a factor sampled uniformly from $U[-1, 1]$. This simulates unreliable supervision signals and is equivalent to injecting label noise. The results show that CAdam produces smoother trajectories and avoids erratic updates, outperforming Adam in terms of stability and robustness under noisy feedback.

Details of both experiments are provided in Appendix C.1. We also include a version of the experiment without noise; the corresponding plot is shown in Figure 6.

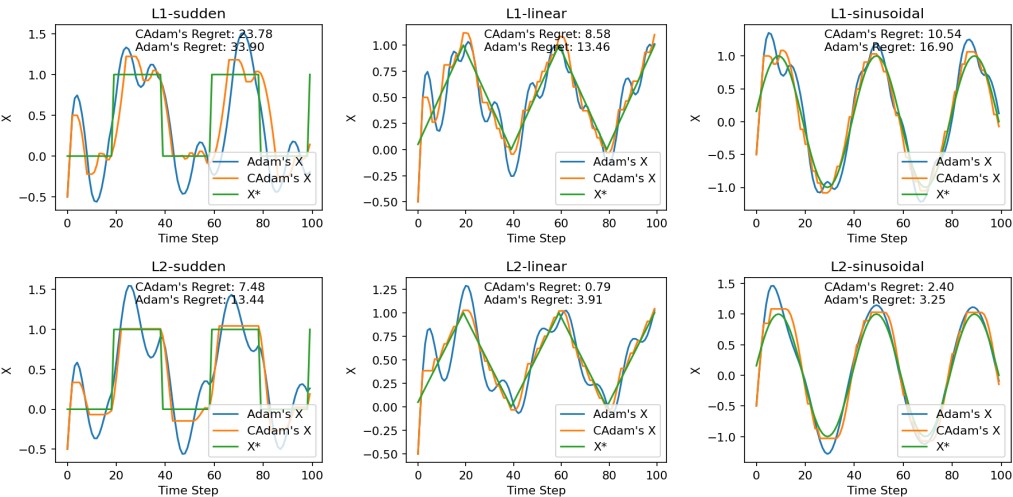

Figure 2: Trajectory of Adam (top row) and CAdam (bottom row) under different distribution shifts. Adam's $X$ and CAdam's $X$ denote the locations of the optimization trajectories for Adam and CAdam, respectively, while $X^*$ represents the location of the optimal solution. CAdam shows superior adaptability to distribution shifts.

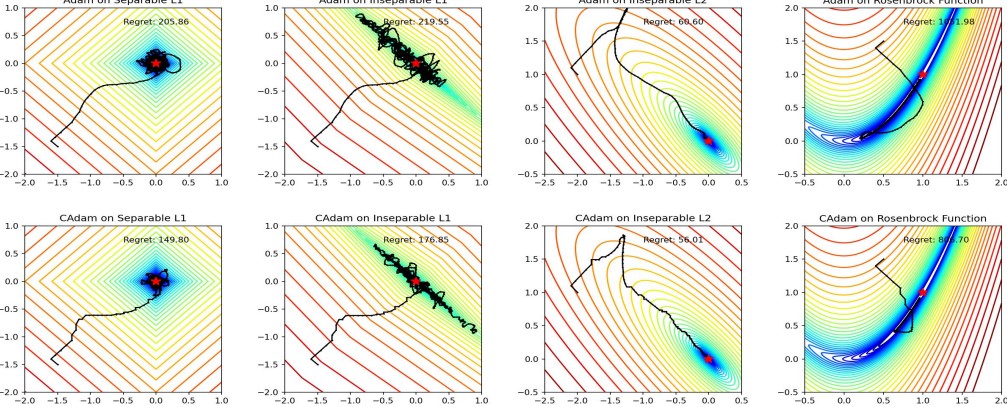

Figure 3: Trajectory of Adam (top row) and CAdam (bottom row) under noisy conditions on four different optimization landscapes.

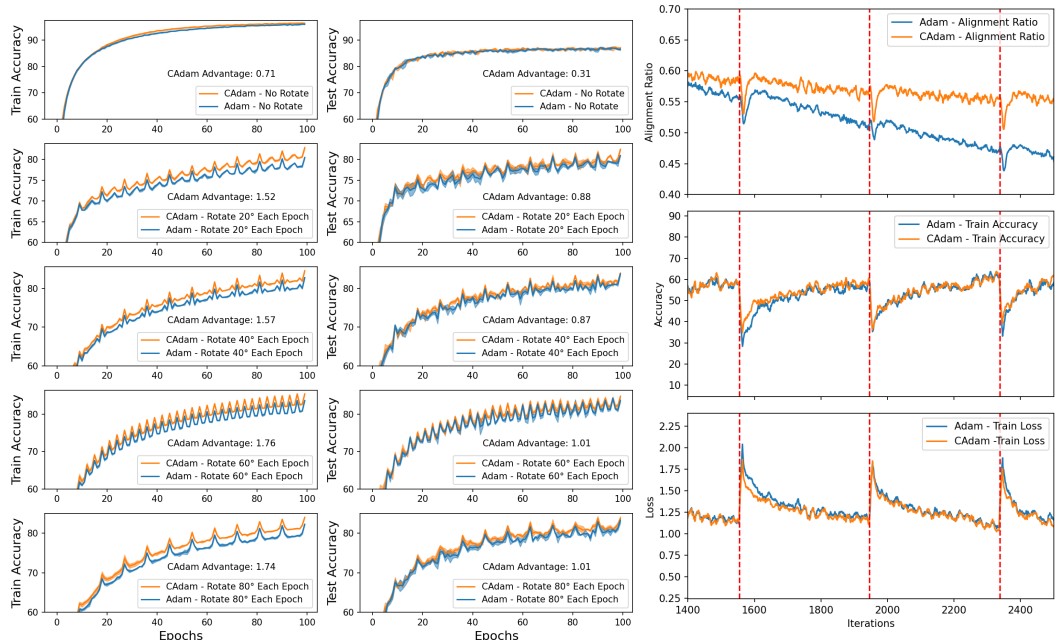

Figure 4: **(Left)** Performance of CAdam and Adam under different rotation speeds corresponding to sudden distribution shift. **(Right)** A detailed view at a $60°$ rotation between steps 1400 to 2300.

## 3.2 CNN ON IMAGE CLASSIFICATION

We perform experiments using the VGG(Simonyan and Zisserman, 2014), ResNet(He et al., 2016) and DenseNet(Huang et al., 2017) on the CIFAR-10 dataset to evaluate the effectiveness of CAdam in handling distribution shifts and noise. We synthesize three experimental conditions: (1) sudden distribution changes, (2) continuous distribution shifts, and (3) added noise to the samples. The hyperparameters for these experiments are provided in Section C.3. In this section, we present only the results for VGG, and the results for ResNet and DenseNet are included in the appendix.

**Sudden Distribution Shift** To simulate sudden changes in data distribution, we rotate the images by a specific angle at the start of each epoch, relative to the previous epoch, as illustrated in Figure 4. CAdam consistently outperforms Adam across varying rotation speeds, with a more significant performance gap compared to the non-rotated condition. We define the *alignment ratio* as:

$$AR = \frac{\text{\# of parameters where } m_t \cdot g_t > 0}{\text{\# of total parameters}}$$

A closer inspection in Figure 4 (Right) reveals that, during the rotation (indicated by the red dashed line), the alignment ratio decreases, resulting in fewer parameters being updated, followed by a gradual recovery. Correspondingly, the accuracy declines and subsequently improves, while the loss increases before decreasing. Notably, during these shifts, CAdam's accuracy drops more slowly and recovers faster than Adam's, indicating its superior adaptability to new data distributions.

**Continuous Distribution Shifts** In contrast to sudden distribution changes, we also tested the scenario where the data distribution changes continuously. Specifically, we simulated this by rotating the data distribution at each iteration by an angle. The results, shown in Figure 5 (Left), indicate that as the rotation speed increases, the advantage of CAdam over Adam becomes more pronounced.

**Noisy Samples** To assess the optimizer's robustness to label noise, we inject noise by randomly selecting a proportion of batches in each epoch (resampled every epoch) and replacing their labels with random values. The results are shown in Figure 5 (Right). As the noise level increases, CAdam becomes more conservative—its consistency score drops, leading it to update fewer parameters per iteration. Despite this cautious behavior, CAdam consistently outperforms Adam in terms of test accuracy. Notably, even under 40% label noise, CAdam achieves comparable accuracy to Adam's performance in the noise-free setting by the end of training.

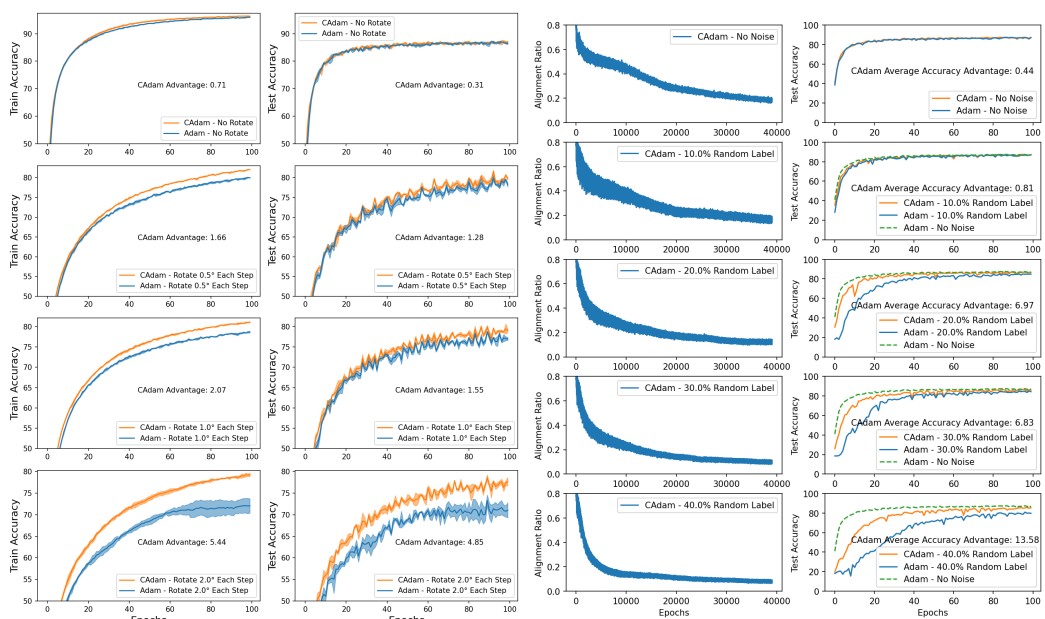

Figure 5: **(Left)** Performance of CAdam and Adam under continuous distribution shifts with different rotation speeds. **(Right)** The effect of adding noise to the samples.

## 3.3 PUBLIC ADVERTISEMENT DATASET

**Experiment Setting** To evaluate the effectiveness of the proposed CAdam optimizer, we conducted experiments using various models on the Criteo-x4-001 dataset(Jean-Baptiste Tien, 2014). This dataset contains feature values and click feedback for millions of display ads and is commonly used to benchmark algorithms for click-through rate (CTR) prediction(Zhu et al., 2021). To simulate a real-world online learning scenario, we trained the models on data up to each timestamp in a single epoch(Fukushima et al., 2020). The other training details are provided in Appendix C.4. We benchmarked CAdam and other popular optimizers, including Adam(Kingma and Ba, 2015), AMS-Grad(Reddi et al., 2018), Yogi(Zaheer et al., 2018a), RAdam(Liu et al., 2019), and AdaBelief(Zhuang et al., 2020), on various models such as DeepFM(77M)(Guo et al., 2017), WideDeep(77M)(Cheng et al., 2016), DNN(74M)(Covington et al., 2016), PNN(79M)(Qu et al., 2016), and DCN(74M)(Wang et al., 2017). The performance of these optimizers was evaluated using the Area Under the Curve (AUC) metric.

**Main Results** The results in Table 1 show that CAdam and its AMSGrad variants outperform other optimizers across different models. While the AMSGrad variants perform better on certain datasets, they do not consistently outperform standard CAdam. Both versions of CAdam generally achieve higher AUC scores than other optimizers, demonstrating their effectiveness in the online learning setting. We note that a 0.05% increase in GAUC, as observed on the Criteo dataset, is nontrivial given that the maximum GAUC difference among models trained with CAdam is only 0.09%. At high baseline accuracy, achieving gains comparable to those obtained through structural model modifications underscores the effectiveness of CAdam.

**Robustness under Noise** To simulate a noisier environment, we introduced noise into the Criteo x4-001 dataset by flipping 1% of the negative training samples to positive. All other settings remained unchanged. The results in Table 2 show that CAdam consistently outperforms Adam in terms of both AUC and the extent of performance drop. This demonstrates CAdam's robustness in handling noisy data.

## 3.4 EXPERIMENT ON REAL-WORLD RECOMMENDATION SYSTEM

Compared to offline experiments, real-world online recommendation systems are significantly more complex. The models we tested range from 8.3 billion to 330 billion parameters—up to 10,000 times larger—and their outputs directly influence user behavior. To evaluate CAdam in this setting, we

Table 1: AUC performance of different optimizers on the Criteo dataset across various models. Results are averaged over three seeds with mean and standard deviation ($\pm$) reported. CAmsGrad denotes the AMSGrad variant of CAdam, which achieves the highest average performance.

| | DeepFM | WideDeep | DNN | PNN | DCN | Avg |
|---|---|---|---|---|---|---|
| Yogi | $80.42_{\pm.006}$ | $80.69_{\pm.005}$ | $80.68_{\pm.014}$ | $80.52_{\pm.014}$ | $80.40_{\pm.020}$ | 80.52 |
| RAdam | $80.84_{\pm.020}$ | $80.91_{\pm.008}$ | $80.89_{\pm.001}$ | $80.96_{\pm.022}$ | $80.90_{\pm.002}$ | 80.90 |
| Adam | $80.87_{\pm.011}$ | $80.90_{\pm.004}$ | $80.89_{\pm.003}$ | $80.90_{\pm.006}$ | $81.05_{\pm.005}$ | 80.92 |
| AdaBelief | $80.84_{\pm.008}$ | $80.90_{\pm.002}$ | $80.88_{\pm.011}$ | $80.89_{\pm.002}$ | $81.02_{\pm.044}$ | 80.91 |
| AdamW | $80.87_{\pm.008}$ | $80.90_{\pm.010}$ | $80.88_{\pm.010}$ | $80.90_{\pm.002}$ | $81.00_{\pm.047}$ | 80.91 |
| AmsGrad | $80.88_{\pm.004}$ | $80.92_{\pm.008}$ | $80.91_{\pm.001}$ | $80.92_{\pm.009}$ | $81.08_{\pm.009}$ | 80.94 |
| CAdam | $80.88_{\pm.008}$ | $80.93_{\pm.004}$ | $80.90_{\pm.002}$ | $80.93_{\pm.006}$ | $81.06_{\pm.009}$ | 80.94 |
| CAmsGrad | $\mathbf{80.90_{\pm.006}}$ | $\mathbf{80.93_{\pm.007}}$ | $\mathbf{80.92_{\pm.005}}$ | $\mathbf{80.94_{\pm.009}}$ | $\mathbf{81.09_{\pm.010}}$ | **80.96** |

Table 2: AUC performance of Adam and CAdam on the Criteo dataset (Noiseless and Noisy versions), averaged over three seeds. $\Delta$ columns show the difference in performance between the Noisy and Noiseless datasets. CAdam generally shows a smaller performance drop.

| | Setting | DeepFM | WideDeep | DNN | PNN | DCN | Avg |
|---|---|---|---|---|---|---|---|
| | Clean | $80.87_{\pm.011}$ | $80.90_{\pm.004}$ | $80.89_{\pm.003}$ | $80.90_{\pm.006}$ | $81.05_{\pm.005}$ | 80.92 |
| Adam | Noisy | $80.51_{\pm.008}$ | $80.47_{\pm.006}$ | $80.48_{\pm.014}$ | $80.66_{\pm.006}$ | $80.51_{\pm.010}$ | 80.53 |
| | $\Delta$ | $-0.36$ | $-0.43$ | $-0.41$ | $-0.23$ | $-0.54$ | $-0.39$ |
| | Clean | $80.88_{\pm.008}$ | $80.93_{\pm.004}$ | $80.90_{\pm.002}$ | $80.93_{\pm.006}$ | $81.06_{\pm.009}$ | 80.94 |
| CAdam | Noisy | $80.81_{\pm.007}$ | $80.79_{\pm.006}$ | $80.78_{\pm.005}$ | $80.96_{\pm.026}$ | $80.77_{\pm.007}$ | 80.82 |
| | $\Delta$ | $-0.08$ | $-0.14$ | $-0.12$ | $+0.04$ | $-0.28$ | $-0.12$ |

conducted 48-hour A/B tests across seven internal production scenarios (2 pre-ranking, 4 recall, and 1 ranking), serving millions of users, against the currently deployed Adam optimizer.

As shown in Table 3, CAdam consistently outperformed Adam in all cases. While the GAUC gains may appear modest, they translate to significant business value at scale. For example, in a system with over 30 million active users, a 0.1% GAUC improvement corresponds to roughly $15 million in annual revenue. Traditional methods—like feature engineering (3–5 engineer-weeks for 0.05% gain) or scaling up models (10x compute for 0.1–0.3% gain)—are costly. In contrast, CAdam offers a plug-and-play optimization improvement, delivering an average 0.3% GAUC lift efficiently and reliably in production. Beyond A/B testing, CAdam **has been deployed** in 16 real-world scenarios involving millions of users and has been running stably in production for over nine months, further validating its long-term robustness and effectiveness in large-scale online systems.

Table 3: GAUC results for Adam and CAdam in internal experiment settings. "Pr" denotes pre-ranking, "Rec" represents recall, and "Rk" indicates ranking. CAdam consistently outperforms Adam.

| | Pr 1 | Pr 2 | Rec 1 | Rec 2 | Rec 3 | Rec 4 | Rk 1 | Average |
|---|---|---|---|---|---|---|---|---|
| Adam | 87.41% | 82.89% | 90.18% | 82.41% | 84.57% | 85.39% | 88.52% | 85.34% |
| CAdam | 87.61% | 83.28% | 90.43% | 82.61% | 85.06% | 85.49% | 88.74% | 85.64% |
| Impr. | 0.20% | 0.39% | 0.25% | 0.20% | 0.49% | 0.10% | 0.22% | 0.30% |

## 3.5 ABLATION STUDY: EFFECT OF THE CONFIDENCE MECHANISM

To further understand the role of the confidence mechanism—computed from the interaction between the momentum term $m_t$ and the gradient $g_t$—we conducted ablation experiments to examine whether its benefits are independent of Adam's second-moment term $v_t$. Specifically, we tested confidence-augmented variants of SGDM and AMSGrad, which we refer to as CSGDM and CAmsGrad, respectively. Both retain the same momentum formulation $m_t$ as Adam, but SGDM does not use a second-moment estimate, while AMSGrad employs a non-decreasing version of $v_t$. This allows us to isolate the effect of the confidence mechanism from the influence of second-moment adaptation.

We evaluate these variants using the same settings described in the previous section, including **image classification** tasks (VGG network on: (1) *Sudden Distribution Shift*—80° random rotation per epoch; (2) *Continuous Distribution Shift*—2° incremental rotation per step; and (3) *Noise*—40% randomly corrupted labels) and a **recommendation system** setting (Criteo dataset, averaged over DeepFM, WideDeep, DNN, PNN, and DCN models). We use average test accuracy for image classification and GAUC for the recommendation task.

As shown in Table 4, both CSGDM and CAmsGrad outperform their vanilla counterparts, suggesting that the confidence mechanism itself contributes meaningfully to performance improvements. These results indicate that the mechanism could potentially benefit a broader class of momentum-based optimizers, such as Lion(Chen et al., 2023)—a promising direction for future work.

Table 4: Ablation results comparing optimizers with and without the confidence mechanism on image classification (VGG under distribution shift and label noise) and recommendation tasks (Criteo). Adding the confidence mechanism consistently improves performance.

|            | SGDM  | CSGDM | Adam  | CAdam | AmsGrad | CAmsGrad |
|------------|-------|-------|-------|-------|---------|----------|
| Sud. shift | 74.83 | 75.99 | 75.07 | 76.05 | 75.00   | 75.78    |
| Cont. shift| 67.19 | 69.08 | 64.61 | 69.13 | 61.22   | 66.35    |
| Noise      | 76.38 | 77.61 | 73.86 | 77.20 | 74.28   | 77.20    |
| Rec. avg.  | 77.16 | 77.71 | 80.92 | 80.94 | 80.94   | 80.96    |

## 3.6 ADDITIONAL EXPERIMENT

We further benchmark CAdam across diverse challenging tasks, including robustness evaluations on CIFAR-100(Krizhevsky and Hinton, 2009) under distribution shifts and label noise, Tiny-Imagenet-C(Hendrycks and Dietterich, 2019), language modeling with GPT-2(Radford et al., 2019), reinforcement learning with PPO(Schulman et al., 2017), and numerical optimization experiments. Detailed results are provided in Appendix B.

## 4 CONVERGENCE ANALYSIS

We follow the theoretical framework established by Li et al. (2023) to analyze the convergence of the CAdam optimizer for non-convex optimization problems of the form:

$$\min_x f(x), \tag{1}$$

where $f$ is a non-convex objective function satisfying rather relaxed smoothness conditions.

**Assumption 4.1.** The objective function $f$ is differentiable and closed within its open domain $\text{dom}(f) \subset \mathbb{R}^d$ and is bounded from below (namely, $\Delta_1 := f(x_1) - f^* < \infty$).

**Assumption 4.2.** The objective function $f$ is $(\rho, L_0, L_\rho)$-smooth with $0 \le \rho < 2$:

$$\|\nabla^2 f(x)\| \le L_0 + L_\rho \|\nabla f(x)\|^2, \quad a.e. \tag{2}$$

Note that, in comparison to the stringent settings employed in the early proofs for the online learning scenario Kingma and Ba (2015); Reddi et al. (2018), the aforementioned assumptions are relatively mild. Specifically, the objective function $f$ is neither convex nor $L$-smooth. The $(L_0, L_\rho)$-smooth function is highly prevalent and encompasses a wide range of classic objective functions (refer to Appendix B.1 in Li et al. (2023) for more examples). Let $r := \min\left\{\frac{1}{5L_\rho G^\rho}, \frac{1}{5(L_0^{\rho-1} L_\rho)^{1/\rho}}\right\}$, $L := 3L_0 + 4L_\rho G^\rho$. and $c_1, c_2$ denote some small enough numerical constants while $C_1, C_2$ denote some large enough ones. We can prove the following convergence rate of the CAdam optimizer:

**Theorem 4.3.** *Suppose Assumption 4.1 and 4.2 hold. Denote $\iota := \log(2/\delta)$ for any $0 < \delta < 1$, and let $G$ be a constant satisfying $G \ge \max\left\{2\epsilon, \sqrt{C_1 \Delta_1 L_0}, (C_1 \Delta_1 L_\rho)^{\frac{1}{2-\rho}}\right\}$. Choose*

$$0 \le \beta_2 \le 1, \quad 1 - \beta_1 \le \min\left\{1, \frac{c_1 \epsilon \gamma^2}{G\sqrt{\iota}}\right\}, \quad \alpha \le c_2 \min\left\{\frac{r\epsilon}{G}, \frac{\epsilon^{3/2}(1-\beta_1)}{L\sqrt{G}}\right\}.$$

*After $T = \max\left\{\frac{1}{(1-\beta_1)^2}, \frac{C_2 \Delta_1 G}{\alpha \gamma^2}\right\}$ CAdam iterations, we have $\frac{1}{T}\sum_{t=1}^T \|\nabla f(x_t)\|^2 \le \gamma^2$ with probability at least $1 - \delta$.*

By fixing $\gamma$, Theorem 4.3 suggests that $T \geq \max\{1/(1-\beta_1)^2, \mathcal{O}(\alpha\gamma^{-2})\}$ is almost sufficient to achieve converged results. When a large $\beta_1$ is selected and the learning rate $\alpha$ is set to the magnitude of $\mathcal{O}(\gamma^2)$, it indicates a convergence rate of $T = \mathcal{O}(\gamma^{-4})$, comparable to the best known convergence rate Li et al. (2023); Wang et al. (2024). Remark that Theorem 4.3 is not intended to demonstrate the superiority of CAdam in terms of convergence. Rather, it is to show that the modification introduced by CAdam does not compromise the ease of use characteristic of Adam-style optimizers. Therefore, CAdam can indeed serve as a substitute for the Adam or AdamW optimizer without raising any theoretical concerns.

## 5 RELATED WORK

**Adam Extensions** Adam is one of the most widely used optimizers, and researchers have proposed various modifications to address its limitations. AMSGrad (Reddi et al., 2018) addresses Adam's non-convergence issue by introducing a maximum operation in the denominator of the update rule. RAdam (Liu et al., 2019) incorporates a rectification term to reduce the variance caused by adaptive learning rates in the early stages of training, effectively combining the benefits of both adaptive and non-adaptive methods. AdamW (Loshchilov, 2017) separates weight decay from the gradient update, improving regularization. Yogi (Zaheer et al., 2018b) modifies the learning rate using a different update rule for the second moment to enhance stability. AdaBelief (Zhuang et al., 2020) refines the second-moment estimation by focusing on the deviation of the gradient from its exponential moving average rather than the squared gradient. This allows the step size to adapt based on the "belief" in the current gradient direction, resulting in faster convergence and improved generalization. Our method, CAdam, similarly leverages the consistency between the gradient and momentum for adjustments. However, it preserves the original update structure of Adam and considers the sign (directional consistency) between momentum and gradient, rather than their value deviation, leading to better performance under distribution shifts and in noisy environments. Very recently, Liang et al. (2024) also explores masking updates based on confidence. While both approaches share a similar intuition, our method focuses on robustness in online learning, while their work focuses on accelerating convergence.

**Adapting to Distributional Changes in Online Learning** In online learning scenarios, models encounter data streams where the underlying distribution can shift over time, a phenomenon known as concept drift (Lu et al., 2018). Adapting to these changes is essential for maintaining model performance. One common strategy is to use sliding windows or forgetting mechanisms (Bifet and Gavalda, 2007), which focus updates on the most recent data. Ensemble methods (Street and Kim, 2001) maintains a collection of models trained on different time segments and combine their predictions to adapt to emerging patterns. Adaptive learning algorithms, such as Online Gradient Descent (Zinkevich, 2003), dynamically adjust the learning rate or model parameters based on environmental feedback. Meta-learning approaches (Finn et al., 2017) aim to develop models that can quickly adapt to new tasks or distributions with minimal updates. Additionally, (Viniski et al., 2021) demonstrated that streaming-based recommender systems outperform batch methods in supermarket data, particularly in handling concept drifts and cold start scenarios.

## 6 CONCLUSION

In this paper, we identified key limitations of Adam in online learning—namely, its slow response to distribution shifts and vulnerability to noisy updates—and proposed CAdam, a simple yet effective variant that incorporates a confidence-based update mechanism. CAdam retains Adam's original structure and convergence rate, enabling seamless replacement in real-world systems.

Through numerical experiments, public benchmarks, and large-scale A/B tests in industrial recommendation systems, we showed that CAdam consistently improves robustness and adaptability while requiring no additional tuning. Future work may explore extending this confidence mechanism to other optimizers and learning settings, and its application to a broader range of machine learning models and real-time systems. Overall, CAdam offers a practical and principled step toward more reliable optimization in dynamic environments.

## 7 REPRODUCIBILITY STATEMENT

To facilitate reproducibility, we provide the core implementation in the anonymous supplementary material, covering all key steps for training, evaluation, and result reproduction.

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

## A   ADDITIONAL RELATED WORK

**Online Learning**   Online learning is commonly formalized via online convex optimization, where algorithms seek low regret against a (possibly adversarial) sequence of losses. Foundational methods include Online Gradient Descent (OGD) (Zinkevich, 2003), Follow-The-Regularized-Leader and its proximal variant (FTRL-Proximal) (McMahan et al., 2013), per-coordinate adaptive methods such as AdaGrad (Duchi et al., 2011), and second-order schemes like Online Newton Step (ONS) (Hazan et al., 2007). In large-scale recommender systems, linear/wide components often use FTRL-style updates for sparsity and feature churn, while deep components are trained with AdaGrad/Adam (Cheng et al., 2016). Our optimizer, CAdam, remains in the SGD/Adam family but adds a confidence gate that suppresses updates when the gradient misaligns with momentum, complementing adaptive/dynamic-regret approaches and orthogonal robustness tricks such as gradient clipping.

**Robustness to Noisy Data**   General methods for noise robustness include robust loss functions (Ghosh et al., 2017), which modify the objective function to reduce sensitivity to mislabeled or corrupted data; regularization techniques (Srivastava et al., 2014), which prevent overfitting by introducing noise during training; and noise-aware algorithms (Gutmann and Hyvärinen, 2010), which explicitly model noise distributions to improve learning. In recommendation systems, enhancing robustness against noisy data is crucial and is typically addressed through two main strategies: *detect and correct* and *detect and remove*. *Detect and correct* methods, such as AutoDenoise (Ge et al., 2023) and Dual Training Error-based Correction (DTEC) (Panagiotakis et al., 2021), identify noisy inputs and adjust them to improve model accuracy by leveraging mechanisms like validation sets or dual error perspectives. Conversely, *detect and remove* approaches eliminate unreliable data using techniques such as outlier detection with statistical models (Xu et al., 2022) or semantic coherence assessments (Saia et al., 2016) to cleanse user profiles. While these strategies can effectively enhance recommendation quality, they often require explicit design and customization for specific models or tasks, limiting their general applicability.

## B   ADDITIONAL EXPERIMENTS

### B.1   EXPERIMENT ON REINFORCEMENT LEARNING

To demonstrate robustness in intrinsically non-stationary scenarios, we evaluated the performance of PPO using Adam variants on the discrete-control environment CartPole-v1 and the continuous-control environment Pendulum-v1, conducting extensive hyperparameter searches over 5 seeds each. As summarized in Table 5, CAdam variants outperform their Adam counterparts.

Table 5: Performance comparison in reinforcement learning environments.

| Environment | CAdam | CAdamW | Adam | AdamW |
|---|---|---|---|---|
| CartPole-v1 | **404.47** | 401.78 | 385.23 | 365.62 |
| Pendulum-v1 | -272.76 | **-225.09** | -306.86 | -357.24 |

### B.2   EXPERIMENT ON LANGUAGE MODELING

We evaluated optimizer performance in large-scale language modeling by training GPT-2 (124M) on the OpenWebText dataset. Extensive hyperparameter searches revealed that although validation perplexities were comparable across optimizers (Table 6), CAdam and CAdamW demonstrated superior training stability, particularly at higher learning rates.

Table 6: GPT-2 Validation Perplexity on OpenWebText

| Optimizer | Best Validation PPL |
|---|---|
| AdamW | **3.02134** |
| CAdamW | 3.02148 |
| Adam | 3.09908 |
| CAdam | 3.09257 |

### B.3 Experiment with More Optimizers

To ensure a comprehensive baseline comparison, we evaluated additional optimizers including AdamNoMomentum, AMSGrad, and Lion on CIFAR datasets under sudden distribution shifts and label noise conditions. Results (Table 7) show CAdam effectively balances robustness across diverse challenging scenarios.

Table 7: Comprehensive Optimizer Comparison on CIFAR

| Dataset | Scenario | CAdam | CAdamW | AdamNoMomentum | AMSGrad | AdamW | Adam | Lion |
|---------|----------|-------|--------|----------------|---------|-------|------|------|
| CIFAR-100 | Rotate 60° | **58.34%** | 56.85% | 57.90% | 56.49% | 56.42% | 55.99% | 49.31% |
| CIFAR-100 | 40% Label Noise | 57.29% | **59.67%** | 45.29% | 54.43% | 55.88% | 53.26% | 57.11% |
| CIFAR-10 | Rotate 60° | **88.11%** | 87.19% | 86.18% | 85.67% | 85.99% | 86.01% | 81.03% |
| CIFAR-10 | 40% Label Noise | 78.96% | 84.43% | 76.62% | 76.55% | 78.52% | 75.95% | **86.32%** |

### B.4 Corrupted Tiny-ImageNet Experiments

To evaluate robustness under image corruptions, we report results on the Corrupted Tiny-ImageNet (Tiny-ImageNet-C) benchmark, following the ImageNet-C protocol (mean top-1 accuracy averaged over corruption types and severities).

Table 8: Tiny-ImageNet-C: Mean top-1 accuracy (%).

| Optimizer | Accuracy (%) | Gap vs Adam |
|-----------|--------------|-------------|
| CAdamW | **49.36** | +0.56 |
| Adam | 48.80 | 0.00 |

### B.5 Numerical Experiments

We performed numerical experiments to benchmark CAdam's performance extensively against other optimizers, ensuring fairness through optimizer-specific hyperparameter tuning.

**Win Rate Against Other Optimizers**    Table 9 summarizes win rates and mean regret of CAdam against other optimizers in non-stationary and noisy gradient settings. CAdam consistently demonstrates superior performance.

Table 9: Numerical Optimization Performance

| Optimizer | Win Rate (Non-stationary) | Win Rate (Noisy) | Mean Regret (Non-stationary) | Mean Regret (Noisy) |
|-----------|---------------------------|------------------|------------------------------|---------------------|
| CAdam | - | - | **14.161** | 152.963 |
| Adam | 0.972 | 0.726 | 20.035 | **150.364** |
| AMSGrad | 0.916 | 0.853 | 19.311 | 212.777 |
| AdamNoMomentum | 0.638 | 0.927 | 17.322 | 236.414 |
| FTRL | 0.694 | 0.392 | 20.462 | 208.638 |
| Lion | 1.000 | 0.917 | 34.722 | 266.865 |

**Trajectory Examples in noise-free enviroment**    Figure 6 illustrate how both optimizers perform in a noise-free environment.

### B.6 Experiment on Resnet and Densenet

We perform experiments on cifar10 setting using Resnet(He et al., 2016) and Densenet(Huang et al., 2017) to further illustrate the effectiveness of CAdam on different architecture.

## C Experiment Detail and Hyperparameters

### C.1 Numerical Experiment Detail

**Distribution Change**    To illustrate the different behaviours of Adam and CAdam under distribution shifts, we designed three types of distribution changes for both L1 and L2 loss functions: (1)

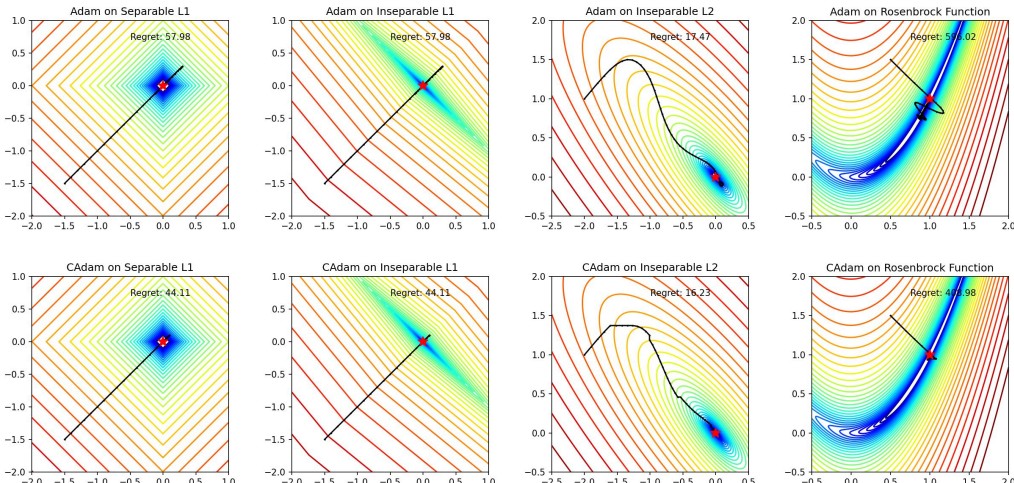

Figure 6: Performance of Adam (top row) and CAdam (bottom row) on four different optimization landscapes without noise: (Left to Right) separable L1 loss, inseparable L1 loss, inseparable L2 loss, and Rosenbrock function. This comparison highlights the natural behavior of both optimizers in a noise-free environment.

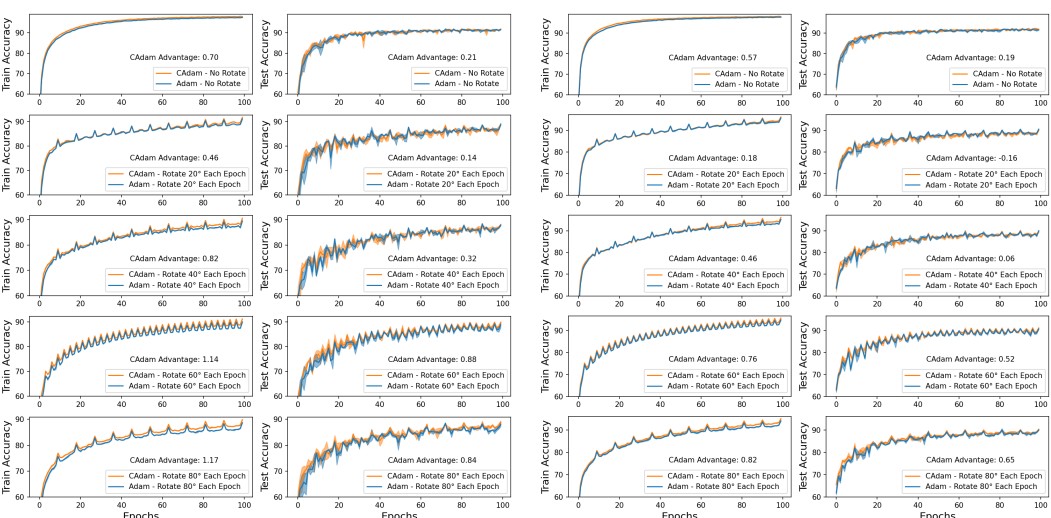

Figure 7: Performance of CAdam and Adam under different rotation speeds corresponding to sudden distribution shift. The results for ResNet are shown on the left, while those for DenseNet are presented on the right.

*Sudden* change, where the minimum shifts abruptly at regular intervals; (2) *Linear* change, where the minimum moves at a constant speed; and (3) *Sinusoidal* change, where the minimum oscillates following a sine function, resulting in variable speed over time.

The loss functions are defined as:

$$L(x, t) = \begin{cases} |x - x^*(t)|, & \text{L1 loss,} \\ (x - x^*(t))^2, & \text{L2 loss,} \end{cases}$$

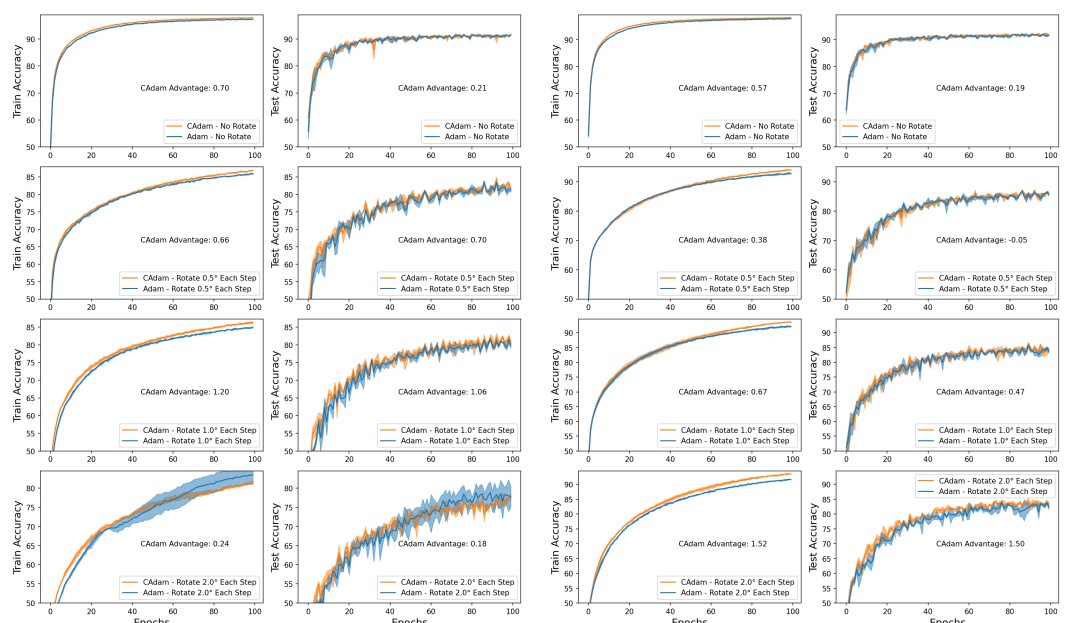

Figure 8: Performance of CAdam and Adam under different rotation speeds corresponding to continuous distribution shift. The results for ResNet are shown on the left, while those for DenseNet are presented on the right.

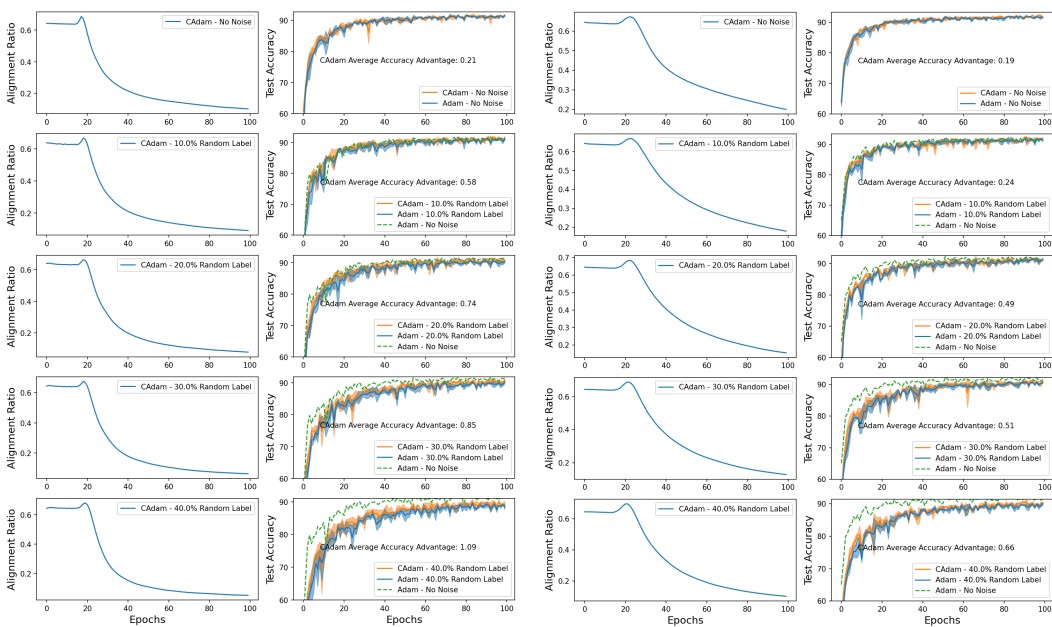

Figure 9: Performance of CAdam and Adam under noisy data. The results for Resnet are shown on the left, while those for Densenet are presented on the right.

where $x^*(t)$ represents the position of the minimum at time $t$ and is defined based on the type of distribution change:

$$x^*(t) = \begin{cases} \lfloor \frac{t}{T} \rfloor \mod 2, & \textit{sudden change}, \\ \frac{t}{T}, & \textit{linear change}, \\ \sin\left(\frac{2\pi t}{T}\right), & \textit{sinusoidal change}. \end{cases}$$

The results of these experiments are presented in Figure 2. Across different loss functions and distribution changes, CAdam closely follows the trajectory of the minimum point, being less affected by incorrect momentum, exhibiting lower regret and demonstrating its superior ability to adapt to shifting distributions.

**Noisy Samples**  To compare Adam and CAdam in noisy environments, we conducted experiments on four different optimization 2-d landscapes:

1. Separable L1 Loss: $f_1(x, y) = |x| + |y|$.

2. Inseparable L1 Loss: $f_2(x, y) = |x + y| + \frac{|x-y|}{10}$.

3. Inseparable L2 Loss: $f_3(x, y) = (x + y)^2 + \frac{(x-y)^2}{10}$.

4. Rosenbrock Function: $f_4(x, y) = (a - x)^2 + b(y - x^2)^2$, where $a = 1$ and $b = 100$.

To simulate noise in the gradients, we applied a random mask to each dimension of the gradient with a 50% probability using the same random seed across different optimizers:

$$\nabla_{\text{noisy}}(x, y) = \begin{cases} \nabla f(x, y) \cdot U(-1, 1), & \text{with } p = 0.5, \\ \nabla f(x, y), & \text{otherwise}, \end{cases}$$

### C.2  Numerical Experiment Hyperparameters

**Distribution Shift**  For the distribution shift experiments, we used the following hyperparameters: a cycle length of 40, a learning rate $\alpha = 0.5$, exponential decay rates for the first and second moment estimates $\beta_1 = 0.9$ and $\beta_2 = 0.999$ respectively, and a small constant $\epsilon = 1 \times 10^{-8}$ to prevent division by zero. The number of time steps was set to $T = 100$.

**Noisy Samples**  For the noisy samples experiments, the hyperparameters were set as follows: a learning rate of 0.1, $\beta_1 = 0.9$, $\beta_2 = 0.999$, $\epsilon = 1 \times 10^{-8}$, and a maximum number of iterations $T = 1500$.

### C.3  Image classification Hyperparameters

For the CNN-based image classification experiments on the CIFAR-10 dataset, we performed hyperparameter selection for Adam from $\{1 \times 10^{-5}, 3 \times 10^{-5}, 1 \times 10^{-4}, 3 \times 10^{-4}, 1 \times 10^{-3}\}$ and applied the optimal learning rate for Adam to both Adam and CAdam. The other hyperparameters were set as follows: $\beta_1 = 0.9$, $\beta_2 = 0.999$, weight decay of 0.0005, and $\epsilon = 1 \times 10^{-8}$.

### C.4  Public Advertisement Experiment Detail

For sparse parameters (e.g., embeddings), we update the optimizer's state only when there is a non-zero gradient for this parameter in the current batch using SparseAdam implementation in Pytorch(Paszke et al., 2019). Due to resource limitations, we performed a grid search over the learning rates for each optimizer and model using the following range: $\{\text{lr\_default}/5, \text{lr\_default}/2, \text{lr\_default}, 2 \times \text{lr\_default}, 5 \times \text{lr\_default}\}$, where lr\_default is the default learning rate specified in the FuxiCTR library. We reported the best performance for each optimizer based on this search. All other hyperparameters were kept the same as those in the FuxiCTR library (Zhu et al., 2021; 2022).

## D  Proof for Non-convex Optimization

Here, we prove that CAdam converges for a non-convex objective function

$$\min\ f(x). \tag{3}$$

We follow the deterministic setting in Li et al. (2023) with the same good convergence rate.

**Theorem D.1** (Restatement of Theorem 4.3). *Suppose Assumption 4.1 and 4.2 hold. Denote $\iota := \log(2/\delta)$ for any $0 < \delta < 1$, and let $G$ be a constant satisfying $G \geq \max\left\{2\epsilon, \sqrt{C_1\Delta_1 L_0}, (C_1\Delta_1 L_\rho)^{\frac{1}{2-\rho}}\right\}$. Choose*

$$0 \leq \beta_2 \leq 1, \quad 1 - \beta_1 \leq \min\left\{1, \frac{c_1\epsilon\gamma^2}{G\sqrt{\iota}}\right\}, \quad \alpha \leq c_2 \min\left\{\frac{r\epsilon}{G}, \frac{\epsilon^{3/2}(1-\beta_1)}{L\sqrt{G}}\right\}.$$

*After $T = \max\left\{\frac{1}{(1-\beta_1)^2}, \frac{C_2\Delta_1 G}{\alpha\gamma^2}\right\}$ CAdam iterations, we have $\frac{1}{T}\sum_{t=1}^{T}\|\nabla f(x_t)\|^2 \leq \gamma^2$ with probability at least $1 - \delta$.*

*Proof.* It is noteworthy that CAdam differs from the original Adam by only incorporating an additional condition to determine whether an update should be performed; that is:

$$x_{t+1} - x_t = -\alpha\hat{m}_{t,\Xi_t}/(\sqrt{\hat{v}_t} + \epsilon). \tag{4}$$

Here, $\hat{m}_{t,\Xi_t}$ indicates the confidence-based moment of which the entries not belonging to $\Xi$ are masked:

$$\hat{m}_{t,\Xi_t} = \begin{cases} \hat{m}_{t,i}, & i \in \Xi_t \\ 0, & \text{else} \end{cases}, \quad \Xi_t := \{i \in [d] : m_{t,i} \cdot g_{t,i} \geq 0\}. \tag{5}$$

Therefore, the proof can be finalized by closely following the original contribution already presented in Li et al. (2023), with the exception of certain lemmas concerning $x_{t+1} - x_t$. In other words, denoted by $\tau$ the first time the sub-optimality gap is strictly greater than a value of $F$ truncated at $T+1$:

$$\tau := \min\{t | f(x_t) - f^* > F\} \wedge (T+1), \tag{6}$$

it is enough to prove that Lemma 5.3 and Lemma C.6 therein hold true for CAdam as well.

**Lemma D.2** (Li et al. (2023) Lemma 5.3). *If $t < \tau$, we have*

$$\|x_{t+1} - x_t\| \leq \alpha D, \quad D := \frac{2G}{\epsilon}.$$

*Proof.* According to Lemma C.2 in Li et al. (2023), we have

$$\|\hat{m}_t\| \leq G. \tag{7}$$

Hence,

$$\begin{aligned} \|x_{t+1} - x_t\| &= \alpha\left\|\frac{\hat{m}_{t,\Xi_t}}{\sqrt{\hat{v}_t} + \epsilon}\right\| \\ &\leq \frac{\alpha}{\epsilon}\|\hat{m}_{t,\Xi_t}\| \\ &\leq \frac{\alpha}{\epsilon}\|\hat{m}_t\| \\ &\leq \frac{\alpha G}{\epsilon} \\ &\leq \frac{2\alpha G}{\epsilon}. \end{aligned} \tag{8}$$

$\square$

**Lemma D.3** (Li et al. (2023) Lemma C.6). *If $t < \tau$, choosing $G \geq \epsilon$ and*

$$\alpha \leq \min\left\{\frac{r}{D}, \frac{\epsilon}{6L}\right\},$$

*we have*

$$f(x_{t+1}) - f(x_t) \leq -\frac{\alpha}{4G}\|\nabla f(x_t)\|^2 + \frac{\alpha}{\epsilon}\|\pi_t\|^2, \tag{9}$$

*where*

$$\pi_t := \hat{m}_t - \nabla f(x_t).$$

*Proof.* According to Lemma C.2 in Li et al. (2023), for $t < \tau$, we have

$$\frac{\alpha I}{2G} \leq \frac{\alpha I}{G + \epsilon} \preceq \underbrace{\text{diag}\left(\frac{\alpha}{\sqrt{\hat{v}_t} + \epsilon}\right)}_{=: H_t} \preceq \frac{\alpha I}{\epsilon}, \tag{10}$$

where $I$ denotes the $d \times d$ identity matrix.

Define the weighted norm $\|x\|_H^2 = x^T H x$ for a positive definite matrix $H$. Then, we have

$$f(x_{t+1}) - f(x_t) \overset{\text{(I)}}{\leq} \langle \nabla f(x_t), x_{t+1} - x_t \rangle + \frac{L}{2}\|x_{t+1} - x_t\|^2$$

$$= \langle \nabla f(x_t), -H_t \hat{m}_{t,\Xi_t} \rangle + \frac{L}{2} \hat{m}_{t,\Xi_t}^T H_t^2 \hat{m}_{t,\Xi_t}$$

$$= \langle \nabla f(x_t), -H_t(\hat{m}_{t,\Xi_t} - \nabla f(x_t) + \nabla f(x_t)) \rangle + \frac{L}{2} \hat{m}_{t,\Xi_t}^T H_t^2 \hat{m}_{t,\Xi_t}$$

$$\overset{\text{(II)}}{\leq} -\|\nabla f(x_t)\|_{H_t}^2 - \left(\nabla f(x_t)\right)^T H_t \underbrace{\left(\hat{m}_{t,\Xi_t} - \nabla f(x_t)\right)}_{=: \pi_{t,\Xi_t}} + \frac{\alpha L}{2\epsilon}\|\hat{m}_{t,\Xi_t}\|_{H_t}^2$$

$$\overset{\text{(III)}}{\leq} -\frac{2}{3}\|\nabla f(x_t)\|_{H_t}^2 + \frac{3}{4}\|\pi_{t,\Xi_t}\|_{H_t}^2 + \frac{\alpha L}{\epsilon}\left(\|\nabla f(x_t)\|_{H_t}^2 + \|\pi_{t,\Xi_t}\|_{H_t}^2\right)$$

$$\overset{\text{(IV)}}{\leq} -\frac{1}{2}\|\nabla f(x_t)\|_{H_t}^2 + \|\pi_{t,\Xi_t}\|_{H_t}^2$$

$$\overset{\text{(V)}}{\leq} -\frac{\alpha}{4G}\|\nabla f(x_t)\|_{H_t}^2 + \frac{\alpha}{\epsilon}\|\pi_t\|^2.$$

The inequalities are justified as follows:

- **(I)** follows from Corollary 5.2 in Li et al. (2023).

- **(II)** uses the fact that $H_t \preceq \alpha I/\epsilon$.

- **(III)** follows from Young's inequality:

$$a^T A b \leq \frac{1}{3}\|a\|_A^2 + \frac{3}{4}\|b\|_A^2$$

and the bound:

$$\|a + b\|_A^2 \leq 2\|a\|_A^2 + 2\|b\|_A^2$$

for any positive semidefinite matrix $A$.

- **(IV)** follows from the condition $\alpha \leq \frac{L}{6L}$.

- **(V)** follows from $\|\pi_{t,\Xi_t}\| \leq \|\pi_t\|$ and the fact that

$$\frac{\alpha I}{2G} \preceq H_t \preceq \frac{\alpha I}{\epsilon}.$$

□

This completes the proof for the deterministic setting by substituting Lemma 5.3 and Lemma C.6 in Li et al. (2023) with Lemma D.2 and Lemma D.3, respectively.

□

# E    CONVERGENCE PROOFS IN THE CONVEX ONLINE LEARNING SETTING

In addition to the proof provided in Appendix D, this section presents a further convergence analysis of CAdam in the convex online learning setting.

Given a stream of objectives $f_t : \mathbb{R}^d \to \mathbb{R}, t = 1, 2, \ldots, T$, online learning aims to minimize the regret w.r.t. the optimum; that is,

$$R_T := \sum_{t=1}^{T} f_t(x_t) - \sum_{t=1}^{T} f_t(x^*), \quad x^* = \underset{x}{\arg\min} \sum_{t=1}^{T} f_t(x). \tag{11}$$

Recall that each update in CAdam can be characterized as follows[1]:

$$m_t = \beta_{1,t} m_{t-1} + (1 - \beta_{1,t}) g_t, \tag{12}$$

$$v_t = \beta_2 v_{t-1} + (1 - \beta_2) g_t^2, \tag{13}$$

$$m_{t,\Xi_t} = \begin{cases} m_{t,i}, & i \in \Xi_t \\ 0, & \text{else} \end{cases}, \tag{14}$$

$$\hat{v}_t = \max(\hat{v}_{t-1}, v_t), \tag{15}$$

$$x_{t+1} = x_t - \alpha_t m_{t,\Xi_t} / \hat{v}_t. \tag{16}$$

where $\Xi_t := \{i \in [d] : m_{t,i} \cdot g_{t,i} \geq 0\}$ indicates the set of active entries at step $t$. For notation clarity, let $x_{t,\Xi}$ be the vector of which the entries not belonging to $\Xi$ are masked. Following the AMSGrad (Reddi et al., 2018), we are to prove that the sequence of points obtained by CAdam satisfies $R_T / T \to 0$ as $T$ increases.

We first introduce three standard assumptions:

**Assumption E.1.** Let $f_t : \mathbb{R}^d \to \mathbb{R}, t = 1, 2, \ldots, T$ be a sequence of convex and differentiable functions with $\|\nabla f_t(x)\|_\infty \leq G_\infty$ for all $t \in [T]$.

**Assumption E.2.** Let $\{m_t\}, \{v_t\}$ be the sequences used in CAdam, $\alpha_t = \alpha / \sqrt{t}, \beta_{1,t} = \beta_1 \lambda^{t-1} < 1, \gamma = \beta_1 / \sqrt{\beta_2} < 1$ for all $t \in [T]$.

**Assumption E.3.** The points involved are within a bounded diameter $D_\infty$; that is, for the optimal point $x^*$ and any points $x_t$ generated by CAdam, it holds $\|x_t - x^*\|_\infty \leq D_\infty / 2$.

We present several essential lemmas in the following. Given that some of these lemmas have been partially established in prior works (Kingma and Ba, 2015; Reddi et al., 2018), we include them here for the sake of completeness.

**Lemma E.4.** *For a convex and differentiable function $f : \mathbb{R}^d \to \mathbb{R}$, we have*

$$f(x) - f(y) \leq \langle \nabla f(x), x - y \rangle. \tag{17}$$

**Lemma E.5.** *Under Assumption E.1 and E.2, we have*

$$\begin{aligned}
\left\langle g_{t,\Xi_t}, x_{t,\Xi_t} - x^*_{\Xi_t} \right\rangle \leq \ & \frac{1}{2\alpha_t(1-\beta_{1,t})} \left( \|V_t^{1/4}(x_{t,\Xi_t} - x^*_{\Xi_t})\|^2 - \|V_t^{1/4}(x_{t+1,\Xi_t} - x^*_{\Xi_t})\|^2 \right) \\
& + \frac{\beta_{1,t}}{2\alpha_t(1-\beta_{1,t})} \|V_t^{1/4}(x_t - x^*)\|^2 \\
& + \frac{\alpha_t}{2(1-\beta_{1,t})} \|V_t^{-1/4} m_t\|^2 + \frac{\alpha_t \beta_{1,t}}{2(1-\beta_{1,t})} \|V_t^{-1/4} m_{t-1}\|^2,
\end{aligned} \tag{18}$$

*where $V_t := \mathrm{diag}(\hat{v}_t)$.*

*Proof.* CAdam updates the parameters as follows

$$x_{t+1,\Xi_t} = x_{t,\Xi_t} - \alpha_t m_{t,\Xi_t} / \sqrt{\hat{v}_t} = x_{t,\Xi_t} - \alpha_t V_t^{-1/2} \Big( \beta_{1,t} m_{t-1,\Xi_t} + (1 - \beta_{1,t}) g_{t,\Xi_t} \Big).$$

Subtracting $x^*$ from both sides yields

$$\begin{aligned}
& \|V_t^{1/4}(x_{t+1,\Xi_t} - x^*_{\Xi_t})\|_2^2 \\
= & \|V_t^{1/4}(x_{t,\Xi_t} - x^*_{\Xi_t}) - \alpha_t V_t^{-1/4} m_{t,\Xi_t}\|_2^2 \\
= & \|V_t^{1/4}(x_{t,\Xi_t} - x^*_{\Xi_t})\|_2^2 - 2\langle \alpha_t V_t^{-1/4} m_{t,\Xi_t}, V_t^{1/4}(x_{t,\Xi_t} - x^*_{\Xi_t})\rangle + \|\alpha_t V_t^{-1/4} m_{t,\Xi_t}\|_2^2 \\
= & \|V_t^{1/4}(x_{t,\Xi_t} - x^*_{\Xi_t})\|_2^2 - 2\alpha_t \langle \beta_{1,t} m_{t-1,\Xi_t} + (1 - \beta_{1,t}) g_{t,\Xi_t}, x_{t,\Xi_t} - x^*_{\Xi_t}\rangle + \|\alpha_t V_t^{-1/4} m_{t,\Xi_t}\|_2^2.
\end{aligned}$$

---

[1]Note that we omit the bias corrections for clarity purpose. It is not difficult to modify the proofs to obtain a more general one.

Rearranging the equation gives

$$\left\langle g_{t,\Xi_t}, x_{t,\Xi_t} - x^*_{\Xi_t} \right\rangle = \frac{1}{2\alpha_t(1-\beta_{1,t})}\left(\|V_t^{1/4}(x_{t,\Xi_t} - x^*_{\Xi_t})\|_2^2 - \|V_t^{1/4}(x_{t+1,\Xi_t} - x^*_{\Xi_t})\|_2^2\right)$$

$$- \frac{\beta_{1,t}}{1-\beta_{1,t}}\left\langle m_{t-1,\Xi_t}, x_{t,\Xi_t} - x^*_{\Xi_t} \right\rangle + \frac{\alpha_t}{2(1-\beta_{1,t})}\|V_t^{-1/4}m_{t,\Xi_t}\|_2^2.$$

The results follow from the Cauchy-Schwarz inequality and Young's inequality:

$$-\frac{\beta_{1,t}}{1-\beta_{1,t}}\left\langle m_{t-1,\Xi_t}, x_{t,\Xi_t} - x^*_{\Xi_t} \right\rangle = \frac{\beta_{1,t}}{1-\beta_{1,t}}\left\langle m_{t-1,\Xi_t}, x^*_{\Xi_t} - x_{t,\Xi_t} \right\rangle$$

$$= \frac{\beta_{1,t}}{1-\beta_{1,t}}\left\langle \sqrt{\alpha_t}V_t^{-1/4}m_{t-1,\Xi_t}, \frac{1}{\sqrt{\alpha_t}}V_t^{1/4}(x^*_{\Xi_t} - x_{t,\Xi_t}) \right\rangle$$

$$\leq \frac{\beta_{1,t}}{1-\beta_{1,t}}\left(\sqrt{\alpha_t}\|V_t^{-1/4}m_{t-1,\Xi_t}\| \cdot \frac{1}{\sqrt{\alpha_t}}\|V_t^{1/4}(x^*_{\Xi_t} - x_{t,\Xi_t})\|\right)$$

$$\leq \frac{\beta_{1,t}}{1-\beta_{1,t}}\left(\frac{\alpha_t}{2}\|V_t^{-1/4}m_{t-1,\Xi_t}\|^2 + \frac{1}{2\alpha_t}\|V_t^{1/4}(x_{t,\Xi_t} - x^*_{\Xi_t})\|^2\right)$$

$$\leq \frac{\beta_{1,t}}{1-\beta_{1,t}}\left(\frac{\alpha_t}{2}\|V_t^{-1/4}m_{t-1}\|^2 + \frac{1}{2\alpha_t}\|V_t^{1/4}(x_t - x^*)\|^2\right),$$

and the fact that $\|V_t^{-1/4}m_{t,\Xi_t}\|_2^2 \leq \|V_t^{-1/4}m_t\|_2^2$.

$\square$

**Lemma E.6.** *Under Assumption E.1, E.2, and E.3, we have*

$$\left\langle g_t, x_t - x^* \right\rangle \leq \left\langle g_{t,\Xi}, x_{t,\Xi} - x^*_{\Xi} \right\rangle + \frac{d\beta_1\lambda^{t-1}D_\infty G_\infty}{1-\beta_1}. \tag{19}$$

*Proof.* If the $i$-th entry is not updated at step $t$, i.e., $i \in [d] \setminus \Xi_t$, it can be derived that

$$\left(\beta_{1,t}m_{t-1,i} + (1-\beta_{1,t})g_{t,i}\right) \cdot g_{t,i} \leq 0$$

$$\Rightarrow \left(\beta_{1,t}m_{t-1,i} + (1-\beta_{1,t})g_{t,i}\right) \cdot \text{sgn}(g_{t,i}) \leq 0$$

$$\Rightarrow -\beta_{1,t}|m_{t-1,i}| + (1-\beta_{1,t})|g_{t,i}| \leq 0$$

$$\Rightarrow |g_{t,i}| \leq \frac{\beta_{1,t}}{1-\beta_{1,t}}|m_{t-1,i}|$$

$$\Rightarrow |g_{t,i}| \leq \frac{\beta_{1,t}}{1-\beta_{1,t}}G_\infty \qquad\qquad \leftarrow \text{Assumption } E.1$$

$$\Rightarrow |g_{t,i}| \leq \frac{\beta_1\lambda^{t-1}}{1-\beta_1}G_\infty, \quad i \in [d] \setminus \Xi_t. \qquad \leftarrow \text{Assumption } E.2$$

With Assumption E.3, it immediately yields the desired inequality that

$$\left\langle g_t, x_t - x^* \right\rangle = \left\langle g_{t,\Xi}, x_{t,\Xi} - x^*_{\Xi} \right\rangle + \left\langle g_{t,[d]\setminus\Xi}, x_{t,[d]\setminus\Xi} - x^*_{[d]\setminus\Xi} \right\rangle$$

$$\leq \left\langle g_{t,\Xi}, x_{t,\Xi} - x^*_{\Xi} \right\rangle + \sum_{i=1}^d \frac{\beta_1\lambda^{t-1}D_\infty G_\infty}{1-\beta_1}.$$

$\square$

**Lemma E.7.** *Given Assumption E.1, E.2, and E.3, we have*

$$\sum_{t\in[T]} \frac{\beta_{1,t}}{2\alpha_t(1-\beta_{1,t})}\|V_t^{1/4}(x_t - x^*)\|^2 \leq \frac{dD_\infty^2 G_\infty}{2\alpha(1-\beta_1)(1-\lambda)^2}. \tag{20}$$

*Proof.*

$$\sum_{t\in[T]}\frac{\beta_{1,t}}{2\alpha_t(1-\beta_{1,t})}\|V_t^{1/4}(x_t-x^*)\|^2$$

$$\leq\frac{1}{2\alpha(1-\beta_1)}\sum_{t\in[T]}\sqrt{t}\lambda^{t-1}\|V_t^{1/4}(x_t-x^*)\|^2$$

$$\leq\frac{G_\infty}{2\alpha(1-\beta_1)}\sum_{t\in[T]}\sqrt{t}\lambda^{t-1}\|x_t-x^*\|^2 \qquad\qquad \leftarrow \text{Assumption } E.1$$

$$\leq\frac{dD_\infty^2 G_\infty}{2\alpha(1-\beta_1)}\sum_{t\in[T]}\sqrt{t}\lambda^{t-1} \qquad\qquad\qquad \leftarrow \text{Assumption } E.3$$

$$\leq\frac{dD_\infty^2 G_\infty}{2\alpha(1-\beta_1)}\sum_{t\in[T]}\lambda^{t-1}t$$

$$\leq\frac{dD_\infty^2 G_\infty}{2\alpha(1-\beta_1)}\frac{1}{(1-\lambda)^2}.$$

$\square$

**Lemma E.8** (Reddi et al. (2018) Lemma2). *Under Assumption E.2, we have*

$$\sum_{t\in[T]}\alpha_t\|V_t^{-1/4}m_t\|^2 \leq \frac{\alpha dG_\infty}{(1-\gamma)(1-\beta_1)\sqrt{1-\beta_2}}\sqrt{T}, \qquad (21)$$

*where $\gamma := \beta_1/\sqrt{\beta_2}$.*

We are ready to prove the final results now. Concretely, Theorem 4.3 is a straightfoward corollary of the following conclusion.

**Theorem E.9.** *Under the Assumption E.1, E.2, and E.3, the regret is converged with*

$$R_T \leq \frac{dD_\infty^2 G_\infty\sqrt{T}}{2\alpha(1-\beta_1)} + \frac{d(2\alpha+D_\infty)D_\infty G_\infty}{2\alpha(1-\beta_1)(1-\lambda)^2} + \frac{\alpha dG_\infty\sqrt{T}}{(1-\gamma)(1-\beta_1)^2\sqrt{1-\beta_2}}. \qquad (22)$$

*Proof.* Based on Lemma E.4, Lemma E.5, and Lemma E.6, the regret can be firstly bounded by

$$R_T = \sum_{t\in[T]}(f_t(x_t)-f_t(x^*)) \leq \sum_{t\in[T]}\langle g_t, x_t-x^*\rangle$$

$$\leq \sum_{t\in[T]}\langle g_{t,\Xi_t}, x_{t,\Xi_t}-x_{\Xi_t}^*\rangle + \sum_{t\in[T]}\frac{d\beta_1\lambda^{t-1}D_\infty G_\infty}{1-\beta_1}$$

$$\leq \underbrace{\sum_{t\in[T]}\frac{1}{2\alpha_t(1-\beta_{1,t})}\Big(\|V_t^{1/4}(x_{t,\Xi_t}-x_{\Xi_t}^*)\|^2 - \|V_t^{1/4}(x_{t+1,\Xi_t}-x_{\Xi_t}^*)\|^2\Big)}_{①}$$

$$+ \underbrace{\sum_{t\in[T]}\frac{\beta_{1,t}}{2\alpha_t(1-\beta_{1,t})}\|V_t^{1/4}(x_t-x^*)\|^2}_{②} + \underbrace{\sum_{t\in[T]}\frac{\alpha_t}{2(1-\beta_{1,t})}\|V_t^{-1/4}m_t\|^2}_{③}$$

$$+ \underbrace{\sum_{t\in[T]}\frac{\alpha_t\beta_{1,t}}{2(1-\beta_{1,t})}\|V_t^{-1/4}m_{t-1}\|^2}_{④} + \underbrace{\sum_{t\in[T]}\frac{d\beta_1\lambda^{t-1}D_\infty G_\infty}{1-\beta_1}}_{⑤}.$$

Let us address each term in turn. For the first term, we are to separately bound each entry and the results follows from the summation. For the $i$-th entry, let $\mathcal{T}_+^i = [t : i \in \bar{\Xi}_t]$ be a sequence collecting

all steps that $x_i$ is succesfully updated, and $\tilde{t}_k \in \mathcal{T}_+^i$ be the $k$-th element of $\mathcal{T}_+^i$. For simplicity, we will omit the superscript without ambiguity.

$$
①_i = \sum_{t=\tilde{t}_1}^{\tilde{t}_{|\mathcal{T}_+|}} \frac{1}{2\alpha_t(1-\beta_{1,t})} \Big( (\hat{v}_{t,i}^{1/4}(x_{t,i}-x_i^*))^2 - (\hat{v}_{t,i}^{1/4}(x_{t+1,i}-x_i^*))^2 \Big)
$$

$$
\leq \frac{\hat{v}_{\tilde{t}_1,i}^{1/2}(x_{\tilde{t}_1,i}-x_i^*)^2}{2\alpha_{\tilde{t}_1}(1-\beta_1)} + \frac{1}{2}\sum_{t=\tilde{t}_2}^{\tilde{t}_{|\mathcal{T}_+|}} \Big[ \frac{\hat{v}_{t,i}^{1/2}(x_{t,i}-x_i^*)^2}{\alpha_t(1-\beta_{1,t})} - \frac{\hat{v}_{t-1,i}^{1/2}(x_{t,i}-x_i^*)^2}{\alpha_{t-1}(1-\beta_{1,t-1})} \Big]
$$

$$
= \frac{\hat{v}_{\tilde{t}_1,i}^{1/2}(x_{\tilde{t}_1,i}-x_i^*)^2}{2\alpha_{\tilde{t}_1}(1-\beta_1)} + \frac{1}{2}\sum_{t=\tilde{t}_2}^{\tilde{t}_{|\mathcal{T}_+|}} \Big[ \frac{\hat{v}_{t,i}^{1/2}(x_{t,i}-x_i^*)^2}{\alpha_t(1-\beta_{1,t-1})} \underbrace{- \frac{\hat{v}_{t,i}^{1/2}(x_{t,i}-x_i^*)^2}{\alpha_t(1-\beta_{1,t-1})} + \frac{\hat{v}_{t,i}^{1/2}(x_{t,i}-x_i^*)^2}{\alpha_t(1-\beta_{1,t})}}_{\leq 0}
$$

$$
- \frac{\hat{v}_{t-1,i}^{1/2}(x_{t,i}-x_i^*)^2}{\alpha_{t-1}(1-\beta_{1,t-1})} \Big]
$$

$$
\leq \frac{\hat{v}_{\tilde{t}_1,i}^{1/2}(x_{\tilde{t}_1,i}-x_i^*)^2}{2\alpha_{\tilde{t}_1}(1-\beta_1)} + \frac{1}{2}\sum_{t=\tilde{t}_2}^{\tilde{t}_{|\mathcal{T}_+|}} \underbrace{\frac{1}{1-\beta_{1,t-1}}}_{\leq 1/(1-\beta_1)} \underbrace{\Big[ \frac{\hat{v}_{t,i}^{1/2}(x_{t,i}-x_i^*)^2}{\alpha_t} - \frac{\hat{v}_{t-1,i}^{1/2}(x_{t,i}-x_i^*)^2}{\alpha_{t-1}} \Big]}_{\geq 0 \text{ by } \hat{v}_{t,i} \geq \hat{v}_{t-1,i}}
$$

$$
\leq \frac{\hat{v}_{\tilde{t}_1,i}^{1/2}(x_{\tilde{t}_1,i}-x_i^*)^2}{2\alpha_{\tilde{t}_1}(1-\beta_1)} + \frac{D_\infty^2}{2(1-\beta_1)} \sum_{t=\tilde{t}_2}^{\tilde{t}_{|\mathcal{T}_+|}} \Big[ \frac{\hat{v}_{t,i}^{1/2}}{\alpha_t} - \frac{\hat{v}_{t-1,i}^{1/2}}{\alpha_{t-1}} \Big] \qquad \leftarrow \text{Assumption E.3}
$$

$$
= \frac{\hat{v}_{\tilde{t}_1,i}^{1/2}(x_{\tilde{t}_1,i}-x_i^*)^2}{2\alpha_{\tilde{t}_1}(1-\beta_1)} + \frac{D_\infty^2}{2(1-\beta_1)} \Big[ \frac{\hat{v}_{\tilde{t}_{|\mathcal{T}_+|},i}^{1/2}}{\alpha_{\tilde{t}_{|\mathcal{T}_+|}}} - \frac{\hat{v}_{\tilde{t}_1,i}^{1/2}}{\alpha_{\tilde{t}_1}} \Big]
$$

$$
\leq \frac{D_\infty^2}{2(1-\beta_1)} \frac{\hat{v}_{\tilde{t}_{|\mathcal{T}_+|},i}^{1/2}}{\alpha_{\tilde{t}_{|\mathcal{T}_+|}}} \leq \frac{D_\infty^2 G_\infty \sqrt{T}}{2\alpha(1-\beta_1)}.
$$

Hence,

$$
① = \sum_{i\in[d]} ①_i \leq \frac{d D_\infty^2 G_\infty \sqrt{T}}{2\alpha(1-\beta_1)}. \tag{23}
$$

$$
② = \sum_{t\in[T]} \frac{\beta_{1,t}}{2\alpha_t(1-\beta_{1,t})} \|V_t^{1/4}(x_t-x^*)\|^2 \leq \frac{d D_\infty^2 G_\infty}{2\alpha(1-\beta_1)(1-\lambda)^2} \qquad \leftarrow \text{Lemma E.7.}
$$

$$
③ = \sum_{t\in[T]} \frac{\alpha_t}{2(1-\beta_{1,t})} \|V_t^{-1/4}m_t\|^2 \leq \frac{1}{2(1-\beta_1)} \sum_{t\in[T]} \alpha_t \|V_t^{-1/4}m_t\|^2
$$

$$
\leq \frac{\alpha d G_\infty \sqrt{T}}{2(1-\gamma)(1-\beta_1)^2\sqrt{1-\beta_2}}. \qquad \leftarrow \text{Lemma E.8}
$$

$$
④ = \sum_{t\in[T]} \frac{\alpha_t\beta_{1,t}}{2(1-\beta_{1,t})} \|V_t^{-1/4}m_{t-1}\|^2 \leq \frac{1}{2(1-\beta_1)} \sum_{t\in[T]} \alpha_t \|V_{t-1}^{-1/4}m_{t-1}\|^2
$$

$$
\leq \frac{1}{2(1-\beta_1)} \sum_{t\in[T]} \alpha_{t-1} \|V_{t-1}^{-1/4}m_{t-1}\|^2 = \frac{1}{2(1-\beta_1)} \sum_{t\in[T-1]} \alpha_t \|V_t^{-1/4}m_t\|^2
$$

$$
\leq \frac{\alpha d G_\infty \sqrt{T}}{2(1-\gamma)(1-\beta_1)^2\sqrt{1-\beta_2}}. \qquad \leftarrow \text{Lemma E.8}
$$

$$⑤ = \sum_{t \in [T]} \frac{d\beta_1 \lambda^{t-1} D_\infty G_\infty}{1 - \beta_1} = \frac{d\beta_1 D_\infty G_\infty}{1 - \beta_1} \sum_{t \in [T]} \lambda^{t-1} \leq \frac{dD_\infty G_\infty}{(1 - \beta_1)(1 - \lambda)^2}.$$

Finally, we have

$$R_T \leq \frac{dD_\infty^2 G_\infty \sqrt{T}}{2\alpha(1 - \beta_1)} + \frac{d(2\alpha + D_\infty)D_\infty G_\infty}{2\alpha(1 - \beta_1)(1 - \lambda)^2} + \frac{\alpha d G_\infty \sqrt{T}}{(1 - \gamma)(1 - \beta_1)^2 \sqrt{1 - \beta_2}}.$$

$\square$

## F  LIMITATION

While CAdam demonstrates strong empirical performance and is supported by theoretical guarantees, a few limitations remain. First, our convergence analysis is currently established under a deterministic setting. Extending this analysis to stochastic optimization—more common in practical applications—is an important direction for future work. Second, while our theory shows that CAdam achieves the same convergence rate as Adam, we do not provide a formal guarantee that it is strictly better. In fact, establishing strict superiority in convergence complexity is notoriously difficult for optimizers of this class, and may not fully reflect the practical robustness advantages demonstrated in our experiments.

## G  COMPUTE RESOURCES

All experiments were conducted on an internal GPU cluster equipped with 1024 H20 GPUs (comparable to NVIDIA A100 80GB in compute capability). Each image classification experiment required approximately 1 H20 GPU-hour, with the full suite of image experiments totaling around 120 H20 GPU-hours. The Criteo advertisement experiments consumed approximately 560 H20 GPU-hours. Large-scale internal recommendation experiments—including A/B testing across multiple online scenarios—used over 2,000 H20 GPU-hours.

The compute estimates above reflect only the finalized experiments reported in the paper. Additional resources were also consumed during preliminary hyperparameter tuning, architecture ablations, and failed runs, which are not included in the totals above.

## H  USE OF LLMs

In this work, Large Language Models (LLMs) were utilized as an assistive tool to enhance productivity and clarity. Specifically, their application was twofold: first, to aid in the generation and debugging of code snippets, thereby improving software development efficiency; and second, to refine the language, grammar, and style of this paper. This latter use was employed to ensure the academic rigor and readability of the text, addressing potential linguistic imperfections as the authors are non-native English speakers. The core concepts, experimental design, and scientific contributions presented herein are entirely the work of the authors.

