# OpenReview forum: "CAdam: Confidence-Based Optimization for Online Learning"
_ICLR.cc/2026/Conference — Submitted to ICLR 2026_

### Official Review · Reviewer_XKfG · 2025-10-14

**Soundness:** 2
**Presentation:** 3
**Contribution:** 1
**Rating:** 4
**Confidence:** 5

**Summary:**

This paper proposes CAdam, a confidence-based optimizer addressing Adam’s key limitations in online learning—slow adaptation to data distribution shifts (due to outdated momentum) and vulnerability to data noise (inability to distinguish noise from valid signals). CAdam’s core mechanism evaluates the consistency between momentum ($m_t$) and current gradient ($g_t$) for each parameter: it updates per Adam’s original formula if consistent, pauses updates to observe subsequent iterations if not, thus differentiating real distribution shifts (sustained gradient direction changes) from random noise (temporary gradient fluctuations). Validated via numerical optimization, CIFAR image classification (handling shifts/noise), Criteo ad dataset experiments, and large-scale online recommendation system A/B tests. Theoretically, it matches Adam’s convergence rate under mild assumptions but has limitations.

**Strengths:**

The study has notable strengths: first, its concise, practical confidence mechanism—no extra hyperparameters, serving as a drop-in replacement for Adam—directly solves the key issue of distinguishing shifts from noise in online learning. Second, its comprehensive, persuasive validation covers offline scenarios (numerical, image classification, ad datasets) and industrial online A/B tests (long-term stable operation), fully proving generalization and practicality.

**Weaknesses:**

1. The sensitivity analysis of β₁ and β₂ is missing. For your algorithm, this is critical because β values affect the calculation of the directional inconsistency between the current gradient ($g_t$) and historical momentum ($m_t$) (i.e., $m_t \cdot g_t \leq 0$).
2. The AUC improvement in the offline experiments (Table 1) is too small. Typically, changes at the ten-thousandth level are hard to prove significant—especially with only three repeated experiments. The gap is even within 0.03%, and the standard deviation is large, making it difficult to demonstrate effectiveness.
3. The GAUC improvement in online experiments is at the thousandth level, which I consider significant. However, in A/B tests, it is common to report improvements in business metrics rather than AUC (as inconsistencies may exist between offline and online performance). I need to know the actual improvements in business metrics (e.g., impressions, clicks, duration) and whether there is a trade-off between different metrics.
4. The method is somewhat overly simple. I understand that it distinguishes "distribution shifts" from "noisy samples" via a sliding window-like mechanism (β-controlled decay). This distinction requires detailed elaboration through experimental analysis. Additionally, if distribution shifts occur sharply, the sliding window may not complete its update before another significant shift happens, so more detailed differentiation is needed.
5. Distribution shifts in advertising scenarios are usually not drastic, but they are more severe in content recommendation (e.g., short video recommendation, where user interest shifts faster than in e-commerce or commercialization). Constructing artificial distribution shifts in offline datasets hardly proves rationality. It is recommended to add content recommendation datasets (e.g., KuaiRand) and use real labels sorted by time to verify whether the method can truly alleviate shift issues.
6. Theoretical limitations: 1) Only convergence rate equivalence to Adam is proven; there is no theoretical guarantee that CAdam is strictly superior in robustness (e.g., noise resistance, distribution shift resistance). 2) Some core assumptions are incompatible with real online scenarios—for example, assuming the objective function has a lower bound and the parameter space has a bounded diameter conflicts with high-dimensional, dynamically changing scenarios like recommendation systems. 3) Non-iid characteristics of online learning (e.g., gradient non-stationarity caused by distribution shifts) are not incorporated, so the theoretical framework cannot support the observed distribution shift resistance in experiments.

**Questions:**

Please see Weaknesses.

---

### Official Review · Reviewer_Si9B · 2025-10-19

**Soundness:** 1
**Presentation:** 2
**Contribution:** 1
**Rating:** 2
**Confidence:** 4

**Summary:**

This paper proposes CAdam (Confidence Adaptive Moment Estimation), a variant of Adam designed for online learning under distribution shifts and noisy feedback. The key modification is a sign-consistency filter: each coordinate of the momentum $m_{t}$ is multiplied by an indicator, meaning that updates are skipped when the current gradient and accumulated momentum disagree in sign. The authors claim this simple heuristic improves robustness to noise and abrupt distributional changes.

CAdam is evaluated on synthetic noise experiments, image classification tasks with rotating distributions, public CTR datasets (Criteo), and live A/B tests in a production recommendation system. The paper also includes a convergence theorem under relaxed smoothness assumptions, following existing study.

**Strengths:**

$\textbf{Clear motivation and simple implementation.}$

The idea of incorporating a “confidence” check based on the agreement between momentum and gradient is intuitive, easy to code, and compatible with existing Adam variants (AMSGrad, AdamW).

---

$\textbf{Readable presentation.}$

The paper is well structured, includes algorithm pseudocode, and connects clearly to known Adam literature.

**Weaknesses:**

$\textbf{Heuristic motivation and lack of rigorous insight.}$

The proposed “confidence mask” is largely an intuitive rule rather than a principled optimization mechanism. Adam’s momentum disagreement with the current gradient is common and often necessary for escaping noise or curvature effects. The paper treats this as an error signal, but in nonconvex stochastic regimes, sign disagreement is expected and not inherently harmful.

---

$\textbf{Potential non-convergence under noise.}$

The method may stall indefinitely in the presence of stochastic gradients.
* Consider a noisy gradient oracle $g_{t} \sim \mathcal{N}(0.1, 1)$. Suppose at an early step we observe an unlucky negative realization $g_{1} = -1$ while the true gradient mean is positive. The CAdam masking rule will zero out this coordinate because $m_{t}g_{t} \leq 0$. Once the momentum is suppressed, subsequent positive gradients will also be muted until the sign consistency is restored, causing the algorithm to “freeze” or drift slowly.
* In contrast, plain Adam continues integrating gradient history; its exponential averaging eventually recovers the correct sign of $m_{t}$. Therefore, CAdam can fail to converge even in simple smooth convex problems when noise occasionally flips gradient signs. The proposed mechanism effectively discards informative updates rather than correcting them.

---

$\textbf{Theoretical claim is unconvincing.}$

The convergence theorem closely reproduces existing results for Adam-type methods without analyzing the masking operator that makes CAdam non-smooth and non-differentiable. No justification is given for how the selective-update rule avoids bias or stalling in expectation. This makes the theoretical section ornamental rather than substantive.

---

$\textbf{Marginal empirical gains and weak statistical support.}$

Reported improvements are on the order of 0.02–0.05\%, often within typical random variation, as demonstrated in Table 1. The qualitative experiments (CIFAR-10) do not establish a consistent or general advantage.

---

$\textbf{Relation to prior work.}$

Recent “cautious” or “masking” optimizers (e.g., Liang et al., 2024) explore nearly identical ideas; the paper provides no comparison against these contemporaneous baselines.

**Questions:**

* In multi-dimensional stochastic problems, can the repeated suppression of coordinates lead to biased parameter trajectories?

---

### Official Review · Reviewer_MUYe · 2025-10-28

**Soundness:** 2
**Presentation:** 3
**Contribution:** 1
**Rating:** 2
**Confidence:** 3

**Summary:**

CAdam introduces a confidence-based rule that stops parameter updates when the gradient and momentum disagree. The idea is simple and useful in practice, and is supported by experiments. However, the method’s originality is limited, and its theoretical part largely reproduces prior work without new insights.

**Strengths:**

- The paper addresses a realistic problem in online learning: Adam often struggles when data distributions shift or when noisy labels appear. This is an important setting in recommender systems and streaming learning, and the motivation is easy to follow.
- The authors report long-term deployment in production, with multiple online scenarios and stable performance improvements. This kind of evidence is rare in optimizer papers and adds credibility to the practical value of the method.

**Weaknesses:**

- The core change is to skip updates when momentum and gradient disagree in sign. This is a small modification that resembles earlier ideas such as AdaBelief or Cautious Optimizers, which also adjust step sizes based on confidence in the current gradient. The paper presents a clear engineering improvement, but not a conceptual breakthrough.
- The convergence proof is almost a direct adaptation of *Li et al. (2023)*, which already established convergence under relaxed smoothness assumptions. The same assumptions and lemmas reappear here, with the masking condition added but not deeply analyzed. As a result, the proof mainly shows that CAdam remains safe to use, rather than explaining why it might perform better.
- Distribution shift and label noise are long-studied problems. Domain adaptation, reweighting, robust loss functions, and noise-tolerant training have well-established techniques. CAdam’s update-masking mechanism can be seen as a heuristic shortcut rather than a principled way to handle either problem.
- When gradient directions fluctuate frequently, the masking rule might silence too many dimensions, slowing convergence or creating bias in parameter updates. The paper does not analyze how often masking occurs or how it changes the effective learning rate over time.

**Questions:**

- Since the theory and assumptions are borrowed from *Li et al. (2023)*, why is that optimizer not included in the experiments? Would CAdam still outperform it?
- How does CAdam compare conceptually and empirically with established methods for handling distribution shift and noisy labels, such as domain adaptation or robust losses?
- Which parts of the proof are genuinely new? For instance, do Lemma D.2 or D.3 differ meaningfully from those in *Li et al.* ?
- Are the observed performance gains statistically significant across different random seeds or hyperparameter settings?
- Does the masking rule create any imbalance across layers in large models, where correlated parameters may require synchronized updates?

---

### Official Review · Reviewer_6T4Z · 2025-10-30

**Soundness:** 3
**Presentation:** 3
**Contribution:** 3
**Rating:** 6
**Confidence:** 3

**Summary:**

This paper introduces CAdam (Confidence Adaptive Moment Estimation), an extension of Adam designed for online learning under non-stationary and noisy data. The core idea is to introduce a confidence mechanism that measures alignment between the momentum $m_t$ and current gradient $g_t$. ​Updates are applied only when $m_t \cdot g_t > 0$, meaning the direction of the accumulated momentum and current gradient are consistent. This selective update rule allows CAdam to:
1. Avoid outdated momentum during distribution shifts, and
2. Suppress noisy gradients that would otherwise destabilize training.

The authors provide both theoretical convergence guarantees (under non-convex and convex online settings) and extensive empirical validation. Experiments span synthetic dynamic systems, image classification (CIFAR-10/100, Tiny-ImageNet-C), language modeling (GPT-2), reinforcement learning (PPO), and large-scale online recommendation (Criteo and industrial A/B tests). Across all these, CAdam consistently outperforms Adam, AMSGrad, and other variants, showing higher robustness to noise and faster adaptation to shifting distributions. Notably, it achieved measurable 0.3% GAUC improvement in production-scale recommendation systems serving millions of users.

**Strengths:**

Originality:
The paper introduces a per‑coordinate confidence gate that updates a parameter only when the sign of the momentum and the current gradient agree, implemented as a one‑line mask $\hat{m}_t \leftarrow \hat{m}_t \odot \mathbb{1}[m_t \odot g_t > 0]$ in Algorithm 1 (line 14). This reframes gradient–momentum alignment as a practical proxy for confidence—an elegant, minimal change that preserves Adam's structure while directly targeting online non‑stationarity and noisy feedback. The intuition is clearly illustrated in Figure 1, which contrasts behavior under distribution shift and label noise.

Quality:
- The empirical suite is broad and convincing; synthetic moving‑optimum and noisy‑landscape studies (Figures 2–3) show faster adaptation and smoother trajectories; CIFAR‑10 under sudden and continuous rotational shifts and injected label noise (Figures 4–5) demonstrates that the method’s advantage widens as shift/noise increases. The paper also introduces an interpretable diagnostic—the alignment ratio AR (definition in p.5)—and shows how it drops at shift points and recovers as the model adapts (Figure 4, right).

- On large‑scale CTR prediction (Criteo‑x4‑001), CAdam/CAmsGrad achieve the best average AUC across five architectures (Table 1), and remain robust when labels are perturbed (smaller performance drop than Adam in Table 2). The paper further validates external impact with seven 48‑hour online A/B tests showing consistent GAUC lifts (average +0.30%) and documents sustained production deployment in 16 scenarios for >9 months (Table 3).

- The confidence mechanism generalizes beyond Adam: ablation results show CSGDM and CAmsGrad outperform their vanilla counterparts on both vision and CTR tasks (Table 4), suggesting transferability to other momentum‑based optimizers. Additional experiments on CIFAR‑100, Tiny‑ImageNet‑C, GPT‑2 language modeling, and PPO reinforcement learning underscore robustness across modalities (§3.6; Tables 5–8). Hyper-parameter protocols and training details are documented in Appendix C, supporting fair comparisons and reproducibility.

- The theory matches the claims; Theorem 4.3 establishes non‑convex convergence under relaxed smoothness without degrading Adam's rate, and Appendix E provides an online convex regret analysis in the AMSGrad style—exactly the guarantees one wants for a "drop‑in" optimizer.

Clarity: Notation and algorithmic steps are succinct (Algorithm 1), figures are task‑aligned (e.g., shift/noise visuals in Figures 2–5), and the alignment ratio offers a transparent, causal readout linking mechanism to observed accuracy/loss dynamics (Figure 4‑right).

Significance: CAdam is hyper-parameter‑free beyond standard Adam settings and functions as a seamless drop‑in replacement, lowering adoption friction in production systems. The paper quantifies real‑world value—e.g., a 0.1% GAUC improvement can translate to substantial revenue at scale—and reports durable production success across many live scenarios (§3.4). The consistency of gains across vision, language modeling, RL, and recommendation indicates the idea captures a general optimization principle rather than a domain‑specific trick.

Overall commendation: This work exemplifies how a simple, principled modification can yield reliable improvements in dynamic, noisy environments. It is easy to implement, well‑diagnosed, theoretically sound, and validated at industrial scale—a combination that makes it both academically meaningful and practically valuable.

**Weaknesses:**

Limited Theoretical Novelty:
- While the paper provides a sound convergence analysis, it largely adapts existing frameworks (e.g., Li et al., 2023) with minimal theoretical innovation. The confidence-based masking mechanism—though intuitively appealing—is described as a binary gating function based on the sign of gradient–momentum alignment, which may oversimplify real-world uncertainty. A more rigorous analysis (e.g., probabilistic confidence modeling or adaptive thresholds) could strengthen the contribution.

- The convergence proofs largely mirror those of Adam with minor extensions; the analysis does not prove improved rates, only equivalence. Also, the key non‑convex proof follows Li et al. (2023) in a deterministic setting. Given the online/noisy motivation, a stochastic‑gradient convergence statement (even with standard bounded‑variance assumptions) would better align with the empirical regime; the authors list this as a limitation (App. F).

Comparison to closely/recent related work:
- The paper briefly notes cautious optimizers that also mask/suppress updates (Liang et al., 2024) but does not include head‑to‑head experiments. Given the conceptual similarity, a direct comparison (same codebases/compute) is important to establish novelty and practical benefit.

- Although comparisons include Adam, AMSGrad, AdaBelief, RAdam, and Lion, the empirical section omits some more recent and closely related methods, such as Cautious Optimizers (Liang et al., 2024) and adaptive gradient confidence approaches (e.g., GradNorm, Trust Ratio variants). Including or discussing these would clarify the originality and relevance of CAdam relative to concurrent work.

Simplistic Confidence Criterion:
- The method assumes that momentum–gradient sign agreement is a sufficient proxy for “confidence.” However, this heuristic might fail when gradients oscillate or in high-dimensional spaces where sign mismatches do not necessarily imply unreliability. Empirical ablations (Table 4, p. 8) confirm benefits but lack analysis of cases where confidence masking could harm learning stability or delay convergence.

Typos:
- page 8, in the definition of $r$, $\frac{1}{5 L_{\rho} G^{\rho}}$ -> $\frac{1}{5 L_{\rho} G^{\rho-1}}$
- page 14, in (B.5), "enviroment" -> "environment"
- page 19, (IV) follows from the condition $\alpha \leq \frac{L}{6L}$ -> $\alpha \leq \frac{\epsilon}{6L}$

**Questions:**

Q1. Relation to "cautious/masked" optimizers - What is the key difference from the Cautious Optimizers (Liang et al., 2024) paper; both use $I(m_t \cdot g_t > 0)$ masking. Can you experimentally compare against Liang et al. (2024) on your CIFAR and Criteo setups? Where does your gate differ in practice (e.g., per‑coordinate vs. global, momentum‑aware vs. gradient‑only)?

Q2. Hyper-parameter robustness - Although claimed to require no tuning, sensitivity analyses (especially to $\beta_1$ and learning rate) could be elaborated. For CIFAR‑10 rotations/noise, I think that using Adam's best LR for both Adam and CAdam (C.3) would be not a fully fair protocol; both optimizers should be tuned over the same grid, or paired Bayesian searches with equal budgets. This could affect the magnitude of reported gaps. Hence, for the CV experiments, what happens if you tune CAdam's LR independently (same grid as Adam's)? Does the relative advantage persist or widen/narrow?

Q3. Stochastic theory - Is there a straightforward extension of Theorem 4.3 under bounded‑variance stochastic gradients, perhaps following the template of Li et al. (2023)/Wang et al. (2024) but incorporating the mask? Even a high‑level theorem statement would strengthen the paper's alignment with its online/noisy claims.

Additional Questions:
- (Distribution Shift) When distribution shifts occur frequently, could the gating mechanism suppress too many updates and slow adaptation? Have you measured adaptation latency (e.g., number of steps until recovery) compared to Adam or AdaBelief?

- (Threshold) Did you experiment with softer confidence thresholds—for example, based on cosine similarity between $m_t$ and $g_t$ —rather than strict sign agreement? If so, how do results compare in terms of stability and convergence?

---

### Meta-Review · Area_Chair_aRZY · 2025-12-22

**Summary:**

The paper proposes CAdam, a confidence-based Adam variant for online learning that addresses distribution shifts and data noise via momentum-gradient sign consistency checks. Based on four reviewers’ feedback, the paper demonstrates practical value with industrial A/B test gains but faces critical gaps in theoretical novelty, comparative analysis, and robustness justification.

Reviewers’ core concerns shaping the decision center on several interrelated issues: the paper’s theoretical analysis largely adapts existing frameworks (e.g., Li et al., 2023) with limited novel contributions, lacking proofs of improved convergence rates and stochastic gradient guarantees that align with its online and noisy learning setting; it fails to include head-to-head experimental comparisons with closely related methods like Cautious Optimizers (Liang et al., 2024) despite conceptual similarities, leaving its originality and practical advantages relative to concurrent work unclear; the confidence mechanism relies on a simplistic binary sign agreement between momentum and gradient as a proxy for confidence, without analyzing potential failure cases such as high-dimensional spaces or gradient oscillations, nor addressing risks of stalling or biased convergence; additionally, offline performance gains are marginal and lack strong statistical support, hyperparameter sensitivity analyses (e.g., for β₁ and β₂) are missing, and online experiments focus solely on GAUC without reporting improvements in business metrics that would validate real-world utility.

**Reviewer Concerns:**

Since no rebuttal was provided by the authors, none of the reviewers’ concerns have been addressed.

**Reviewer Scores:**

Due to the absence of a rebuttal from the authors to address the identified outstanding concerns, all reviewers’ assessments remain rooted in the original submission materials. Without additional clarification, supplementary analysis, or responses to critical questions, there is no basis for any reviewer to revise their initial evaluations of the paper’s contribution, soundness, or presentation. As such, the overall scoring landscape is expected to remain unchanged from the original reviews.

---

### Decision · Program_Chairs · 2026-01-26

Reject